# Rethinking the Power of Graph Canonization in Graph Representation Learning with Stability

**Zehao Dong[1] Muhan Zhang[2] Philip R.O. Payne[3] Michael A Province[4] Carlos Cruchaga[5] Tianyu Zhao[6] Fuhai Li[3,7] Yixin Chen[1,† \*]**
{zehao.dong,prpayne,mprovince,cruchagac,tzhao}@wustl.edu
chen@cse.wustl.edu & muhan@pku.edu.cn

## Abstract

The expressivity of Graph Neural Networks (GNNs) has been studied broadly in recent years to reveal the design principles for more powerful GNNs. Graph canonization is known as a typical approach to distinguish non-isomorphic graphs, yet rarely adopted when developing expressive GNNs. This paper proposes to maximize the expressivity of GNNs by graph canonization, then the power of such GNNs is studies from the perspective of model stability. A stable GNN will map similar graphs to close graph representations in the vectorial space, and the stability of GNNs is critical to generalize their performance to unseen graphs. We theoretically reveal the trade-off of expressivity and stability in graph-canonization-enhanced GNNs. Then we introduce a notion of universal graph canonization as the general solution to address the trade-off and characterize a widely applicable sufficient condition to solve the universal graph canonization. A comprehensive set of experiments demonstrates the effectiveness of the proposed method. In many popular graph benchmark datasets, graph canonization successfully enhances GNNs and provides highly competitive performance, indicating the capability and great potential of proposed method in general graph representation learning. In graph datasets where the sufficient condition holds, GNNs enhanced by universal graph canonization consistently outperform GNN baselines and successfully improve the SOTA performance up to 31%, providing the optimal solution to numerous challenging real-world graph analytical tasks like gene network representation learning in bioinformatics. Our source code is available at https://github.com/zehao-dong/RethinkGraphCanonical.

## 1 Introduction

Graph Neural Networks (GNNs) (Kipf & Welling, 2016; Hamilton et al., 2017; Xu et al., 2019; Velickovic et al., 2018; You et al., 2018; Scarselli et al., 2008; Duvenaud et al., 2015; Gilmer et al., 2017; Zhang et al., 2018) are dominant architectures for modeling the relational structured data. GNNs implicitly leverage the inductive bias in the graph topology by iteratively passing messages between nodes to extract a node embedding that encodes the local substructure, showing great expressivity and scalability to extract representative features for graph learning problems. Due to the superior representation ability and scalability, GNNs have achieved impressive results on various graph-structured data, such as social networks (Monti et al., 2017; Ying et al., 2018), protein networks (Fout, 2017; Zitnik et al., 2018) and circuit networks (Dong et al., 2022a).

Numerous efforts (Xu et al., 2019; Morris et al., 2019) have been put in analyzing the effectiveness of GNNs and indicate that popular GNNs, which mimic the 1-dimensional Weisfeiler-Lehman (1-WL) algorithm/color refinement algorithm (Leman & Weisfeiler, 1968), suffer from the limitations

---
[*1] Department of Computer Science & Engineering, Washington University in St. Louis. [2] Institute for Artificial Intelligence, Peking University. [3] Institute for Informatics, Data Science, and Biostatistics, Washington University School of Medicine [4] Department of Genetics, Washington University School of Medicine [5] Department of Psychiatry, Washington University School of Medicine [6] Department of Radiation Oncology, Washington University School of Medicine [7] Department of Pediatrics, Washington University School of Medicine[†] Corresponding authors.

of 1-WL in the expressive power. Hence, the dominant approach to enhance the performance of GNNs is to improve their expressivity. Informally, recent powerful GNNs achieve this purpose from four perspective. (1) High-order GNNs (Morris et al., 2019; Maron et al., 2019) mimic the higher-order WL tests to improve the expressicity beyond 1-WL; (2) Subgraph-based GNNs (Zhang & Li, 2021; You et al., 2021; Bevilacqua et al., 2021) encode (learnable) local structures (i.e. induced k-hop neighborhoods) of nodes, thus formulating hierarchies of local isomorphism on neighborhood subgraphs to beat 1-WL. (3) GNNs with markings (Papp et al., 2021) slightly perturb the input graphs for multiple times to execute GNNs and then aggregates the final embeddings of these perturbed graphs in each run. (4) Feature augmentation GNNs (Abboud et al., 2020; Lim et al., 2022) add pre-computed/learnt node features to improve the expressivity. Overall, these expressive GNNs are powerful, yet still far from efficiently distinguishing any pair of nonisomorphic graphs. For instance, high-order GNNs and GNNs with markings are not scalable to large-scale graphs due to the space and time complexity, while subgraph-based GNNs have an upper bounded expressicity as there always exists graphs (Papp & Wattenhofer, 2022) distinguishable by 2-WL (3-WL) yet not distinguishable by subgraph-based GNNs.

Graph canonization generates the canonical form of a graph $G$ that is isomorphic to any graph in it's isomorphic group $ISO(G)$, then any two graphs are distinguishable by checking whether their canonical forms are identical. Thus, graph canonization maximizes the discriminating ability of non-isomorphic graphs. As for the concern of the complexity, though determination of the canonical form of a graph is shown to be NP-hard Babai & Luks (1983), canonization tools such as Nauty (McKay & Piperno, 2014) can effectively find the canonical form of a reasonable-sized graph. For instance, Nauty has an average time complexity of $O(n)$, and polynomial-time graph canonization algorithms also exist for graphs of bounded degrees. Hence, we propose to enhance GNNs with graph canonization.

The positional encoding scheme (Vaswani et al., 2017) is introduced in the sequence encoding problem to describe the position of an entity in a sequence so that each position is associated with a unique representation. In recent years, it is widely adopted in graph Transformers (Yan et al., 2020; Dong et al., 2022b), and we resort to this scheme when applying graph canonization in GNNs. For a graph $G$ with node set $V = \{v_1, v_2, ..., v_n\}$, the graph canonization finds a bijective function $\rho(v|G)$ from $V$ onto $\{1, 2, .., n\}$ such that the canonical form of $G$ can be obtained by reordering nodes according to the generated discrete colouring $\{\rho(v_1|G), \rho(v_2|G), ..., \rho(v_n|G)\}$. Since isomorphic graphs have the same canonical form, function $\rho(v|G)$ provides a unique way to label nodes in graphs from the same isomorphic group $ISO(G)$. Thus, our proposed graph-canonization-enhanced GNNs take one-hot encodings of $\{\rho(v_1|G), \rho(v_2|G), ..., \rho(v_n|G)\}$ as nodes' positional encodings, which help to maximize the expressive power in graph classification.

In the design of deep learning models, the generalization behavior (Bartlett et al., 2017) also plays a critical role for competitive performance. Inspired by Wang et al. (2022), we study how well GNNs are generalized to unseen graphs with the model stability. Informally, a stable GNN can map similar graphs to close graph representations in the vectorial space. In the graph canonization problem, similar graphs may have complete different discrete colourings $\{\rho(v_1|G), \rho(v_2|G), ..., \rho(v_n|G)\}$, causing graph representations learnt by GNNs fail to capture the similarity of input graphs when GNNs are enhanced by graph canonization. We theoretically prove that GNNs using graph canonization to generate nodes' positional encodings are unstable. Thus, there is a trade-off between the stability and expressivity in GNNs equipped with the graph canonization as an enhancer.

In the paper, we theoretically prove that the trade-off can not be generally addressed due to the individualization-refinement paradigm used in practical graph canonization tools. Consequently, we develop a notion of universal graph canonization to address the trade-off. Though the universal graph canonization problem is also NP-hard, we characterize a widely applicable sufficient condition when it is solvable through an injective (node labeling) function $\tau(G|\mathbb{G}) : V \to \mathbb{N}$, which utilizes the augmented information across the whole graph dataset $\mathbb{G} = \{G_n | n = 1, 2, ...N\}$ to introduce node asymmetry consistent with subgraph isomorphism in $\mathbb{G}$. That is, for any common subgrpah $G'$ of $G_1$ and $G_2$ in dataset $\mathbb{G}$, the output of $\tau(G_1|\mathbb{G})$ and $\tau(G_2|\mathbb{G})$ are *identical* on subgraph $G'$. Then, GNNs with nodes' positional encodings of $\{\tau(v_1|\mathbb{G}), \tau(v_2|\mathbb{G}), ..., \tau(v_n|\mathbb{G})\}$ is proven to be stable, indicating that GNNs enhanced by universal graph canonization can maximize the expressivity while maintaining the stability.

The key contributions in this paper are as follows: 1) we propose a novel plug-and-play architecture to enhance GNNs by generating nodes' positional encodings with graph canonization algorithms, and theoretically study its' strengths and limitations. The proposed graph-canonization-enhanced GNN (GC-GNN) has the maximized expressivity in distinguishing non-isomorphic graphs, and our theory reveals the trade-off between the expressivity and model stability. 2) We theoretically solve the trade-off by introducing a notion of universal graph canonization, and propose a widely applicable sufficient condition to solve it for numerous graph datasets, including gene networks, brain networks, Bayesian networks, etc. 3) A comprehensive set of experiments demonstrates the effectiveness of proposed method. In popular graph benchmark datasets, GC-GNN successfully enhances GNNs' performance and achieves competitive results. In graph datasets where the universal graph canonization is tractable by our sufficient condition theorem, the GNNs enhanced by the universal graph canonization (UGC-CGNN) consistently outperforms GNN baselines and improves the SOTA performance up to 31% over baselines.

## 1.1 OTHER RELATED WORKS

Graph canonization problem is proven to be NP-hard (Babai & Luks, 1983). Practical canonization tools such as Nauty (McKay & Piperno, 2014) and Bliss (Junttila & Kaski, 2012), are developed to effectively find the canonical form in practice. PATCHY-SAN (Niepert et al., 2016) uses graph canonization tools as a black-box function to determine the global/local node order so that CNNs can be applied to fixed-size patches extracted from nodes' neighborhood. On the other hand, PFGNN (Dupty et al., 2021) train neural architectures with particle filtering to mimic the paradigm of individualization and refinement (McKay & Piperno, 2014; Junttila & Kaski, 2011) in practical graph canonization tools.

## 2 GRAPH CANONIZATION AND STABILITY OF GNNS

In this section, we introduce useful concepts. and then reveal the trade-off between the expressivity and stability in GNNs enhanced by graph canonization. Let $G = (V, E, c)$ denote a colored graph of $n$ nodes, where $V = \{v_1, v_2, ..., v_n\}$ is the set of nodes, $E \in V \times V$ is the set of edges, and the function $c : V \to \mathbb{N}$ associates to each node an integer color (node type). The edge information is always expressed as an adjacency matrix $A \in \{0, 1\}^{n \times n}$ such that $A_{i,j} = 1$ iff edge $(i, j) \in E$.

A *colouring* of $G$ is a surjective function from the node set $V$ onto a set of $k$ integers. In this paper, We use "colouring" to denote both the image of this surjective function (i.e. set of integers). A colouring is discrete if $k = n$, then each cell (i.e. set of nodes with the same image) is a singleton.

### 2.1 GRAPH DISTANCE AND STABLE GNNS

Let $\pi : V \to V$ be a permutation operation on the node set $V$. We denote by $v^\pi$ the image of any node $v \in V$, then $\pi(G)$ (i.e. permutation operation on a graph $G$) generates another colored graph $\pi(G) = G^\pi = (V, E^\pi, c^\pi)$ such that (1) $(v_1^\pi, v_2^\pi) \in E^\pi$ iff $(v_1, v_2) \in E$ (i.e. $A_{v_1, v_2}^\pi = A_{v_1, v_2}$) and (2) $c^\pi(v^\pi) = c(v)$. Each permutation operation $\pi$ associates to a $n$-dimension permutation matrix $P^\pi$ in $\{0, 1\}^{n \times n}$, where each row and each column has only one single 1. All the permutation operation $\pi$ on graphs of $n$ nodes formulate a permutation group $\Pi(n)$.

Given two $n$-size graphs $G_1 = (V_1, E_1, c_1)$, $G_2 = (V_2, E_2, c_2)$ and their corresponding adjacency matrix $A_1$ and $A_2$, there exists a permutation operation $\pi^* \in \Pi(n)$ associates to the permutation matrix $P^*$ that best aligns the graph structures and the node colors (i.e. node features),

$$P^* = argmin_{P \in \Pi} ||A_1 - P A_2 P^T||_F + \sum_i^n c_1(i)! = c_2(\pi(i))$$

The distance between two graphs is thus characterized as $d(G_1, G_2) = ||A_1 - P^* A_2 P^{*T}||_F + \sum_i c_1(i)! = c_2(\pi^*(i))$, which measures the similarity of two graphs of the same size.

Equipped with the notion of graph distance, we characterize the the stability of GNN models. The GNN model provides graph representation vector based on adjacency information and node features, then we express it as $g(A, X)$, where $X$ is a 2-dimension tensor of one-hot features of node colors.

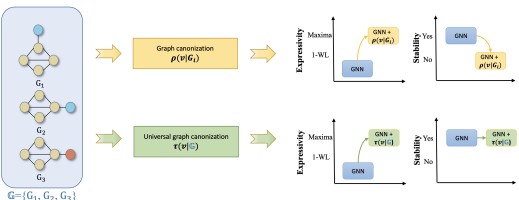

Figure 1: Overview of graph canonization techniques in the design of GNNs. Graph canonization improves the expressivity of GNNs at the cost of stability. Consequently, universal graph canonization is proposed to alleviate the problem .

Unless specified, following theoretical analysis assumes a GNN model consists of several message passing (MP) layers and a sum readout layer.

**Definition 2.1** *(Stability). A GNN model $g$ is claimed to be stable under $X$, if there exists a constant $C > 0$, for any two graphs $G_1$, $G_2$, $g$ satisfies $||g(A_1, X_1) - g(A_2, X_2)||_2 \leq Cd(G_1, G_2)$.*

In analog to Wang et al. (2022), the stability of a GNN model guarantees that similar graphs in the graph distance space will be mapped to similar representations in the vectorial space. Hence, it will provide good generalization ability to apply GNNs on unseen graphs. Overall, the stability and expressivity are two fundamental design principles of GNN architectures. The stability helps to bound the gap between the representations of an unseen testing graph and a similar but different training graph, while the higher expressivity enables GNNs to recognize more graph structures.

## 2.2 GNNs with Graph Canonization

Two colored graphs $G_1$ and $G_2$ of $n$ nodes are isomorphic (denoted by $G_1 \simeq G_2$) if there exists a permutation operation $\pi \in \Pi(n)$ such that $\pi(G_1) = G_2$. The set all colored graphs isomorphic to $G$ forms it's isomorphism class $ISO(G) = \{G^{'}|G^{'} \simeq G\}$, while an automorphism of $G$ is an isomorphism that maps $G$ onto itself.

**Definition 2.2** *(Graph canonization) Graph canonization $f_n$ is a function that maps a colored graph to an isomorphism of the graph itself (a graph in $ISO(G)$) such that $\forall \pi \in \Pi(n)$, $f_n(G) = f_n(\pi(G))$.*

In other words, graph canonization assigns to each coloured graph an isomorphic coloured graph that is a unique representative of its isomorphism class. The order of nodes $v \in f_n(G)$ are denoted as $\rho(v|G)$. Then, the graph canonization provides a unique discrete colouring $\{\rho(v_1|G), \rho(v_2|G), ..., \rho(v_n|G)\}$ as nodes' positions for each isomorphism class $ISO(G)$ to break the symmetry of nodes.

**Lemma 2.3** *Let $P$ be a 2-dimension tensor of one-hot features of the discrete colouring generated by an arbitrary graph canonization algorithm. A GNN model $g$ that takes $P$ as nodes' positional encodings maximizes the expressivity, if graph convolution layers of $g$ are injective.*

**Lemma 2.4** *Let $P$ be a 2-dimension tensor of one-hot features of the discrete colouring generated by a graph canonization algorithm follows the individualization-refinement paradigm. A GNN model $g$ is stable under $X$, and is not stable under $X \oplus P$, where $\oplus$ denotes the concatenation operation.*

We prove lemma 2.3 and lemma 2.4 in Appendix C. Graph canonization algorithms that follow the individualization-refinement paradigm  Grohe et al. (2017) will respect the order of initial node colors/types when breaking ties to introduce the node asymmetry, thus there always exists similar graphs to have complete different discrete colourings as Figure  2 illustrates. These two lemmas indicate the trade-off of expressivity and stability in GNNs enhanced by graph canonization. Then, graph-canonization-enhanced GNNs (GC-GNNs) can distinguish more graph structures in the training data, yet the ability of generalizing well to unseen testing data is deteriorated to some extend. Hence, we study how to overcome this trade-off in next section. Furthermore, since the graph canonization problem is NP-hard, and practical tools like Nauty (McKay & Piperno, 2014) and Bliss  (Junttila & Kaski, 2012) for the problem always follow the individualization-refinement paradigm, the trade-off is not generally solvable in practice.

## 3    UNIVERSAL GRAPH CANONIZATION

In this section, we first introduce the notion of universal graph canonization, and theoretically prove that it can maximize GNNs' expressive power without losing the stability. Then we propose a widely applicable sufficient condition to apply GNNs with universal graph canonization in numerous graphs.

### 3.1    UNIVERSAL GRAPH CANONIZATION

Ideally, the maximally powerful GNNs are expected to be expressive and stable. Thus, our goal is to generate a discrete colouring $\{\tau(v_1), \tau(v_2), ..., \tau(v_n)\}$ that is equivalent to the output colouring $\{\rho(v_1|G), \rho(v_2|G), ..., \rho(v_n|G)\}$ of a general graph canonization algorithm in breaking the node symmetry, while maintaining the stability to provide better generalization performance.

**Definition 3.1** *(Universal graph canonization) Let $\mathbb{G} = \{G_i | i = 1, 2, ..., N\}$ be a graph dataset. A discrete colouring function $\tau(v|\mathbb{G}) : V \to \mathbb{N}$ is claimed as an universal graph canonization for $\mathbb{G}$, if $\forall$ common subgrpah $G'$ of $G_1$, $G_2 \in \mathbb{G}$. $\tau(G_1|\mathbb{G})$ and $\tau(G_2|\mathbb{G})$ are identical in $G'$.*

It is straightforward that the output discrete colouring $\{\tau(v_1|\mathbb{G}), \tau(v_2|\mathbb{G})...\tau(v_n|\mathbb{G})\}$ of the proposed universal graph canonization is equivalent to $\{\rho(v_1|G), \rho(v_2|G), ..., \rho(v_n|G)\}$ of a general graph canonization algorithm in breaking nodes' symmetry and distinguishing non-isomorphic graphs. Both of them will be same for isomorphic graphs and be different for non-isomorphic graphs, as one can get the canonical form of an input graph $G$ by sorting nodes according to the discrete colouring $\{\tau(v_1|\mathbb{G}), \tau(v_2|\mathbb{G})...\tau(v_n|\mathbb{G})\}$.

The main difference between the output discrete colourings $\{\tau(v_1|\mathbb{G}), \tau(v_2|\mathbb{G})...\tau(v_n|\mathbb{G})\}$ and $\{\rho(v_1|G), \rho(v_2|G), ..., \rho(v_n|G)\}$ is their resistance ability to graph perturbations. The graph canonization tools employ the technique of individualization-refinement to sequentially refine a colouring until it is discrete. Then, the output discrete colouring $\{\rho(v_1|G), \rho(v_2|G), ..., \rho(v_n|G)\}$ is heavily dependent on the order of nodes to individualize, which is determined by the symmetry in the graph structure. Consequently, even a small perturbation that breaks the symmetry, such as recoloring a single node, may causes a significant change of the sequence of individualized nodes, resulting in a compete different output colouring. That is, the change of output colouring happens across all nodes regardless of their geometric relation to the position of the perturbated node. Figure 2 provides an example as the illustration. After we recolor the top yellow node as orange (i.e. $G_1 \to G_2$), the output discrete colouring (i.e. numbers on nodes) of left triangle and bottom triangle are completely different, yet these subgrpahs are isomorphic in $G_1$ and $G_2$. In contrast, the universal graph canonization generate the discrete colouring conditioned on whole graph dataset $\mathbb{G}$ to guarantee the color consistency among subgraphs $G'$ of any pair of graphs in $\mathbb{G}$. Thus, the graph perturbation to nodes/edges outside $G'$ will not affects generated colours within $G'$.

**Theorem 3.2** *Let $\tau(v|\mathbb{G})$ be an universal graph canonization for dataset $\mathbb{G}$, and $T$ be the 2-dimensional tensor of one-hot features of discrete colouring $\{\tau(v_1|\mathbb{G}), \tau(v_2|\mathbb{G})...\tau(v_n|\mathbb{G})\}$. A GNN model $g$ under $X \oplus T$ is stable, where $\oplus$ denotes the concatenation operation.*

We prove Theorem 3.2 in Appendix D. This theory guarantees that GNNs enhanced by universal graph canonization (UGC-GNNs) are stable. In addition, since universal graph canonization is equally powerful as the general graph canonizaton in distinguishing non-isomorphic graphs, UGC-GNNs can also maximize the expressive power in graph discrimination.

### 3.2    A SUFFICIENT CONDITION FOR GNNs WITH UNIVERSAL GRAPH CANONIZATION

Having recognized the stability and expressivity of UGC-GNN, we next discuss its' applicability in real-world graph representation learning. The main challenge is to effectively solve the universal graph canonization problem on a given dataset $\mathbb{G}$. Unlike the general graph canonization problem where many practical tools, such as Nauty (McKay & Piperno, 2014) and Bliss (Junttila & Kaski, 2012), are developed to effectively implement the algorithm, there is no existing software to solve the universal graph canonization problem. Furthermore, according to the definition 3.1, this problem is at least as hard as the subgraph isomorphism problem, thereby is NP-hard.

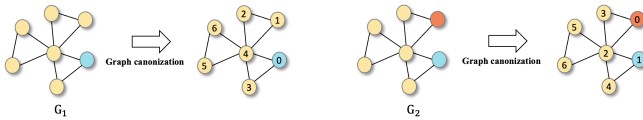

Figure 2: Illustration of the limitation in graph canonization follows the individualization-refinement paradigm. In the example, initial node colors/types follow the order: orange $<$ blue $<$ yellow. Then, two similar graphs $G_1$ and $G_2$ will always have complete different discrete colouring generated by graph canonization algorithms.

**Lemma 3.3** *Let $l(v|\mathbb{G}) : V \to \mathbb{N}$ be an injective function. $l(v|\mathbb{G})$ is a universal graph canonization, if (I) for $\forall v_1, u_1 \in G_1$, $v_2, u_2 \in G_2$, $G_1, G_2 \in \mathbb{G}$ such that $l(v_1|\mathbb{G}) = l(v_2|\mathbb{G})$ and $l(u_1|\mathbb{G}) = l(u_2|\mathbb{G})$, we have $(v_1, u_i) \in E_1 \leftrightarrow (v_2, u_2) \in E_2$, where $\leftrightarrow$ is an equivalence relation. Or (II) $l(v|\mathbb{G})$ is independent of $\mathbb{G}$.*

**Application scenarios** We prove Lemma 3.3 in Appendix E, and this lemma provides a sufficient condition to compute the universal graph canonization. Together with the definition 3.1, universal graph canonization is tractable in numerous scenarios: (1) In gene networks Shuhendler et al. (2010); Dong et al. (2023), each gene appears at most once in each gene network, and the connection of any pair of genes is shared among different gene networks, then $l(v|\mathbb{G})$ can be extracted by lexico-graphically sorting all genes appear in the dataset; (2) In brain graphs Li et al. (2021); Venkataraman et al. (2016), the brain FMRI images are parcellated into ROIs (regions of interests) as nodes, and edges are computed by the functional correlation matrix between ROIs. Since these ROIs can be pre-ordered across all brain images, these pre-orders can be used as the output of the $l(v|\mathbb{G})$. (3) In traffic networks such as METR-LA and Roadnet-CA Li et al. (2017), nodes represent fixed intersec-tions or road endpoints whose pre-defined orders can be treated as the output of $l(v|\mathbb{G})$. (4) Many well-known computer vision datasets such as MNIST and CIFAR10 are also transformed to graph signals on fixed-grid-like structures, where $l(v|\mathbb{G})$ can be defined by lexicographically sorting nodes' grid positions, and these datasets have become benchmarks for graph learning recently Dwivedi et al. (2020); (5) In many DAG learning problems Zhang et al. (2019), such as neural architecture search (NA) and Bayesan structure learning (BN), the function $l(v|\mathbb{G})$ can be obtained by sorting nodes according to their positions in the Hamiltonian path in NA or to their node types in BN.

**Readout layer** Both message passing layers and readout layers contribute to GNN's learning ability in the graph-level prediction. Thus, UGC-GNN also propose to incorporate the discrete coloring $\{l(v_1|\mathbb{G}), l(v_2|\mathbb{G})...l(v_n|\mathbb{G})\}$ in the readout layer to break nodes' symmetry. Specifically, a weighted summation over set of node representation $\{h_v^T | v \in V\}$ is used, where the trainable weight matrices are associated w.r.t the discrete colouring $\{l(v_1|\mathbb{G}), l(v_2|\mathbb{G})...l(v_n|\mathbb{G})\}$,

$$Readout(\{h_v^T | v \in V\}) = \sum_{v \in V} W_{l(v|\mathbb{G})} h_v^T$$

Here, $h_v^T$ is the output representation of node $v$ in the last ($T$-th) message passing layer of UGC-GNN. Thus, UGC-GNN introduces node asymmetry in both graph convolution layers and readout layer.

**Lemma 3.4** *UGC-GNN is permutation invariant.*

We prove Lemma 3.4 in Appendix F. As nodes in a graph have no intrinsic ordering, GNN model should also be permutation invariant to the permutation operations. That is, $\forall P \in \Pi(n)$, we have $g(A, X) = g(PAP^T, AX)$. Machine learning model fail to do so will lead to a waste of training data and computation time. Hence, the proposed UGC-GNN is stable, maximally expressive, and permutation-invariant.

## 4    EXPERIMENTS

In this section, we evaluate the effectiveness of proposed graph-canonization-enhanced GNNs against competitive GNN baselines. Concretely, we first conduct experiments on synthetic datasets and TU datasets to test the general performance of proposed method (GC-GNN). Then we demonstrate the superiority of our UGC-GNN when universal graph canonization is solvable through proposed

sufficient condition, and use challenging gene network datasets, brain graph dataset, DAG datasets as the benchmark in this scenario. Details of datasets are provided in Appendix A.

## 4.1 DATASETS

**Synthetic datasets and TU datasets.** EXP (Abboud et al., 2020) and CSL (Murphy et al., 2019) are two graph isomorphism test datasets where non-isomorphic regular graphs are required to be distinguished. Specifically, the EXP dataset contains 600 pairs of 1-WL-indistinguishable but non-isomorphic graphs, while the CSL dataset contains 150 4-regular graphs from 10 isomorphism groups. TU datasets (Dobson & Doig, 2003; Toivonen et al., 2003) include five graph datasets: PROTEINS, PTC-MR, MUTAG, ENZYMES, D&D. TU datasets are widely used to perform the graph classification task.

**Gene network datasets.** Gene networks are ubiquitous in bioinformatics. In the experiment, we select three gene network datasets: RosMap, Mayo, and Cancer, for the challenging graph classification tasks in bioinformatics. Mayo and RosMap are designed for the Alzheimer's disease (AD) classification (De Jager et al., 2018; Allen et al., 2016); Cancer is designed for the cancer subtype classification. In gene networks, gene expressions are used as node features, while edges between genes are collected from KEGG (Kyoto Encyclopedia of Genes and Genomes) Kanehisa & Goto (2000) based on the physical signaling interactions from documented medical experiments.

**Brain graph datasets.** Brain graphs in the public dataset fMRI ABIDE Di Martino et al. (2014) take as nodes the ROIs (regions of interests) parcellated from brain FMRI images, and use the computed functional correlation matrix between ROIs as edges. The objective is to classify the brain state.

**DAG datasets.** This experiment contains two DAG datasets: NA and BN. In dataset NA, neural architectures generated by the software `ENAS` (Pham et al., 2018), and the corresponding weight-sharing (WS) accuracy on CIFAR-10 (Krizhevsky et al., 2009) is pre-computed for each arcitecture. In dataset BN, Bayesian networks (DAGs) are randomly sampled by the `bnlearn` package (Scutari, 2010). Each DAG is associated with a Bayesian Information Criterion (BIC) score that measures the architecture performance on dataset Asia (Lauritzen & Spiegelhalter, 1988).

## 4.2 BASELINES AND EXPERIMENT CONFIGURATION

The experiment takes three types of baselines: (1) popular GNNs and graph Transformers that achieve top places on OGB leaderboard: GCN (Kipf & Welling, 2016), GIN (Xu et al., 2019), GAT (Velickovic et al., 2018), GraphSAGE (Hamilton et al., 2017), SAN (Kreuzer et al., 2021), Graphormer (Ying et al., 2021); (2) Expressive GNNs beyond 1-WL: NGNN (Zhang & Li, 2021), GCN-RNI (Abboud et al., 2020), PNA (Corso et al., 2020), GINE (Brossard et al., 2020), PPGN (Maron et al., 2019), GraphSNN (Wijesinghe & Wang, 2022) and GNN-AK+ (Zhao et al., 2021); SignNet (Lim et al., 2022), ESAN (Bevilacqua et al., 2021), DropGNN (Papp et al., 2021); (3) Dominant DAG GNNs: GraphRNN (You et al., 2018), S-VAE (Bowman et al., 2016), D-VAE (Zhang et al., 2019), DAGNN(Thost & Chen, 2021).

Gene networks have received considerable attention in bioinformatics and numerous deep learning (DL) models are developed recently to analyze gene networks. Thus, beside above GNN baselines, we also compare our UGC-GNN against powerful DL models in bioinformatics: TransSynergy Liu & Xie (2021), SANEpool Dong et al. (2023), Decagon Zitnik et al. (2018), DimiG Pan & Shen (2019), MLA-GNN Xing et al. (2022). We thoroughly introduce graph-structured data in bioinformatics and their DL baselines in Appendix H to explain why we select above DL models as additional baselines.

We perform 10-fold (or 5-fold) cross validation for robust comparison. All experiments are implemented in the environment of PyTorch using NVIDIA A40 GPUs. Our graph-canonization-based GNNs take backbone GNNs from $\{GIN, GCN, GraphSAGE, GAT\}$. In GNN baselines, the embedding dimension of graph convolution layer is set to be 32. The number of graph convolution layer is selected from the set $\{2, 3, 4\}$. The graph-level readout function is selected from $\{mean, sum, sortpool\}$. In NGNN, we use height-1 rooted subgraphs to avoid the out-of-memory problem in gene network datasets. The experimental settings follow Dong et al. (2022b) on dataset NA/BN and follow Zhang & Li (2021) on TU datasets. The training protocols is composed of the selection of the evaluation rates and training stop rules. Specifically, the learning rate of optimizer picks the best from the set

$\{1e-4, 1e-3, 1e-2\}$; the training process is stopped when the validation metric does not improve further under a patience of 10 epochs.

## 4.3 EVALUATION OF GC-GNN

To test the general performance of GNNs enhanced by graph canonization (i.e. GC-GNN), we conduct experiments to answer following questions: **Q1:** Can GC-GNN empirically achieve the maximal expressive power in distinguishing non-isomorphic graphs? **Q2:** To what extend can GC-GNN improve the predictive performance without ameliorating the model stability?

**Answer to Q1 (Table 1)** Here we start with the first question. In this experiment, we test the expressive power of GC-GNN on two synthetic datasets, EXP and CSL. use GIN and GCN as backbone GNN. As Table 1 shown, for any backbone GNN, GC-GNN can significantly improve the expressive power and distinguishes almost all the 1-WL indistinguishable graphs.

| | EXP | | CSL | |
|---|---|---|---|---|
| Model | Train Accuracy ↑ | Test Accuracy ↑ | Train Accuracy ↑ | Test Accuracy ↑ |
| GCN (backbone) | $0.5000 \pm 0.000$ | $0.5000 \pm 0.000$ | $0.1213 \pm 0.0034$ | $0.0133 \pm 0.0281$ |
| **GC-GNN (GCN)** | $0.9918 \pm 0.0102$ | $0.9267 \pm 0.0858$ | $0.9994 \pm 0.0008$ | $0.9583 \pm 0.0791$ |
| GIN (backbone) | $0.5000 \pm 0.000$ | $0.5000 \pm 0.000$ | $0.1400 \pm 0.0663$ | $0.1157 \pm 0.0081$ |
| **GC-GNN (GIN)** | $1.00 \pm 0.00$ | $0.9975 \pm 0.0056$ | $1.00 \pm 0.00$ | $0.9667 \pm 0.0720$ |

Table 1: Evaluation of expressive power on synthetic datasets for graph isomorphism test.

**Answer to Q2 (Table 2)** Lemma 2.4 demonstrates the limitation of GC-GNN. This experiment empirically show that GC-GNN can still achieve impressive performance when the stability concern is not well addressed. The widely adopted TU datasets are used as the example, and Table 2 presents the results. 1) GC-GNN consistently brings performance gains to GNN backbones by a large margin, and the erformance gain is up to $29.6\%$ 2) Nested GNN has been shown to a powerful plug-and-play framework to improve GNNs' expressivity beyond 1-WL. Compared to this popular subgraph-based GNN, GC-GNN also provides a considerable performance improvement without the additional space/computation cost raised by the subgraph extraction. 3) Compared to recent powerful GNNs like GraphSNN, GNN-AK+, SignNet, ESAN, etc, GC-GNN can still achieve highly competitive performance. For instance, GC-GNN significantly improves the state-of-the-art performance on $D\&D$ to an accuracy above $90\%$.

| | D&D ↑ | MUTAG ↑ | PROTEINS ↑ | PTC-MR ↑ | ENZYMES ↑ |
|---|---|---|---|---|---|
| GCN (backbone) | $71.6 \pm 2.8$ | $73.4 \pm 10.8$ | $71.7 \pm 4.7$ | $56.4 \pm 7.1$ | $27.3 \pm 5.5$ |
| GraphSAGE (backbone) | $71.6 \pm 3.0$ | $74.0 \pm 8.8$ | $71.2 \pm 5.2$ | $57.0 \pm 5.5$ | $30.7 \pm 6.3$ |
| GIN (backbone) | $70.5 \pm 3.6$ | $84.5 \pm 4.9$ | $70.6 \pm 4.3$ | $51.2 \pm 9.2$ | $38.3 \pm 6.4$ |
| GAT (backbone) | $71.0 \pm 4.4$ | $73.9 \pm 10.7$ | $72.0 \pm 3.3$ | $57.0 \pm 7.3$ | $30.2 \pm 4.2$ |
| Nested GCN | $76.3 \pm 3.8$ | $82.9 \pm 11.1$ | $73.3 \pm 4.0$ | $57.3 \pm 7.7$ | $31.2 \pm 6.7$ |
| Nested GraphSAGE | $77.4 \pm 4.2$ | $83.9 \pm 10.7$ | $74.2 \pm 3.7$ | $57.0 \pm 5.9$ | $30.7 \pm 6.3$ |
| Nested GIN | $77.8 \pm 3.9$ | $87.9 \pm 8.2$ | $73.9 \pm 5.1$ | $54.1 \pm 7.7$ | $29.0 \pm 8.0$ |
| Nested GAT | $76.0 \pm 4.4$ | $81.9 \pm 10.2$ | $73.7 \pm 4.8$ | $56.7 \pm 8.1$ | $29.5 \pm 5.7$ |
| GraphSNN | $79.6 \pm 2.9$ | $87.8 \pm 7.6$ | $74.7 \pm 3.9$ | $54.6 \pm 8.5$ | $36.8 \pm 5.1$ |
| GIN-AK+ | $80.3 \pm 3.1$ | $88.5 \pm 8.6$ | $75.2 \pm 4.7$ | $55.1 \pm 9.2$ | $\mathbf{38.5 \pm 6.3}$ |
| SignNet | $78.2 \pm 4.1$ | $88.3 \pm 9.2$ | $75.6 \pm 4.1$ | $64.3 \pm 7.1$ | $37.5 \pm 6.4$ |
| GNN-RNI | $75.8 \pm 3.0$ | $90.4 \pm 7.4$ | $73.5 \pm 4.5$ | $58.2 \pm 6.3$ | $30.7 \pm 5.6$ |
| ESAN | $79.7 \pm 3.8$ | $\mathbf{91.0 \pm 4.8}$ | $75.8 \pm 4.5$ | $65.7 \pm 7.0$ | $37.9 \pm 6.3$ |
| DropGNN | $78.5 \pm 6.0$ | $90.4 \pm 7.0$ | $76.3 \pm 6.1$ | $62.7 \pm 8.4$ | $\mathbf{38.3 \pm 7.1}$ |
| **GC-GNN (GCN)** | $91.3 \pm 9.7$ | $86.2 \pm 9.9$ | $\mathbf{76.7 \pm 5.1}$ | $\mathbf{66.9 \pm 7.1}$ | $37.7 \pm 6.9$ |
| **GC-GNN (GraphSAGE)** | $\mathbf{92.1 \pm 8.1}$ | $89.4 \pm 8.8$ | $75.8 \pm 4.0$ | $62.5 \pm 5.1$ | $35.7 \pm 5.9$ |
| **GC-GNN (GIN)** | $91.4 \pm 8.6$ | $86.2 \pm 6.7$ | $74.4 \pm 3.7$ | $57.2 \pm 7.5$ | $30.7 \pm 4.9$ |
| **GC-GNN (GAT)** | $90.5 \pm 9.0$ | $87.3 \pm 7.5$ | $75.6 \pm 3.8$ | $62.2 \pm 7.0$ | $32.0 \pm 5.8$ |
| **Ave. improvement over backbone** | **28.2%** | **14.6%** | **6.0%** | **12.3%** | **8.9%** |
| **Max. improvement over backbone** | **29.6%** | **20.8 %** | **7.0%** | **18.6%** | **38.1 %** |
| **Ave. improvement over Nested GNN** | **18.7%** | **3.8%** | **2.5%** | **10.5%** | **12.9 %** |
| **Max. improvement over Nested GNN** | **19.7%** | **6.6%** | **4.6%** | **16.8%** | **20.8 %** |

Table 2: Prediction accuracy ($\%$) on TU datasets. Highlighted are the **best** results.

## 4.4 EVALUATION OF UGC-GNN

Though we have shown the huge potential of GC-GNN in enhancing GNNs, the loss of model stability still raises challenges when applied to some graph datasets, especially these with large-scale graphs. For instance, gene networks in datasets Mayo, Rosmap and Cancer contain more than 3000 nodes, then Table 3 shows that GC-GNN with a backbone of GIN even performs worsen than GIN. More similar ablation results are available in Appendix G. Another example is the experiments on OGB datasets. Appendix B shows that GC-GNN can improve base GNNs on OGB datasets (ogbg-molhiv, ogbg-molpcba) to a close performance to ESAN with ED policy, yet they are not comparable to the SOTA performance. These results empirically demonstrate the importance of maintaining GNN's stability and the necessity of designing UGC-GNN.

**The superior representation learning ability. (Table 3 and Table 4)** This experiment evaluates UGC-GNN against powerful GNN baselines on many real-world datasets where Lemma 3.3 can

| Methods | Mayo | | RosMap | | Cancer | | fMRI-ABIDE | |
|---|---|---|---|---|---|---|---|---|
| | Accuracy ↑ | F1 score ↑ | Accuracy ↑ | F1 score ↑ | Accuracy ↑ | F1 score ↑ | Accuracy ↑ | F1 score ↑ |
| GIN | 0.496 ± 0.042 | 0.484 ± 0.036 | 0.471 ± 0.039 | 0482 ± 0.041 | 0.537 ± 0.045 | 0.512 ± 0.047 | 0.592 ± 0.035 | 0.553 ± 0.046 |
| GCN | 0.561 ± 0.049 | 0.535 ± 0.021 | 0.520 ± 0.036 | 0.571 ± 0.032 | 0.593 ± 0.039 | 0.561 ± 0.042 | 0.607 ± 0.031 | 0.566 ± 0.062 |
| GAT | 0.515 ± 0.034 | 0.547 ± 0.021 | 0.491 ± 0.037 | 0.508 ± 0.042 | 0.461 ± 0.039 | 0.532 ± 0.031 | 0.614 ± 0.034 | 0.590 ± 0.052 |
| PNA | 0.551 ± 0.037 | 0.579 ± 0.046 | 0.560 ± 0.035 | 0.584 ± 0.041 | 0.620 ± 0.029 | 0.691 ± 0.033 | 0.613 ± 0.071 | 0.585 ± 0.072 |
| GINE | 0.539 ± 0.041 | 0.571 ± 0.038 | 0.572 ± 0.050 | 0.583 ± 0.041 | 0.629 ± 0.044 | 0.619 ± 0.058 | 0.620 ± 0.029 | 0.592 ± 0.061 |
| NGNN | 0.517 ± 0.033 | 0.504 ± 0.031 | 0.509 ± 0.030 | 0.481 ± 0.042 | 0.516 ± 0.049 | 0.532 ± 0.056 | 0.621 ± 0.051 | 0.557 ± 0.085 |
| PPGN | 0.522 ± 0.021 | 0.510 ± 0.037 | 0.539 ± 0.033 | 0.607 ± 0.037 | 0.499 ± 0.025 | 0.484 ± 0.042 | 0.633 ± 0.042 | 0.600 ± 0.053 |
| GCN-RNI | 0.513 ± 0.027 | 0.501 ± 0.031 | 0.496 ± 0.041 | 0.512 ± 0.037 | 0.521 ± 0.041 | 0.502 ± 0.069 | 0.624 ± 0.021 | 0.598 ± 0.050 |
| SignNet | 0.527 ± 0.035 | 0.514 ± 0.029 | 0.544 ± 0.030 | 0.609 ± 0.037 | 0.572 ± 0.044 | 0.553 ± 0.048 | 0.628 ± 0.041 | 0.597 ± 0.052 |
| ESAN | 0.579 ± 0.035 | 0.605 ± 0.037 | 0.581 ± 0.042 | 0.614 ± 0.040 | OOM | OOM | 0.629 ± 0.058 | 0.596 ± 0.061 |
| DropGNN | 0.533 ± 0.042 | 0.517 ± 0.039 | 0.557 ± 0.045 | 0.571 ± 0.050 | OOM | OOM | 0.619 ± 0.063 | 0.593 ± 0.072 |
| GC-GNN (GIN) | 0.483 ± 0.026 | 0.472 ± 0.031 | 0.486 ± 0.041 | 0.510 ± 0.037 | 0.539 ± 0.041 | 0.532 ± 0.069 | 0.610 ± 0.033 | 0.572 ± 0.056 |
| **UGC-GNN (GIN)** | **0.624± 0.036** | **0.713 ± 0.022** | **0.701± 0.025** | **0.689± 0.019** | **0.714 ± 0.011** | **0.701 ± 0.032** | **0.648± 0.033** | **0.625 ± 0.060** |

Table 3: Experimental results on bioinformatical tasks: gene network datasets and brain graph datasets. Shown is the mean ± s.d. of 10 runs with different random seeds. **Best results** are highlighted.

| Methods | NA | | BN | |
|---|---|---|---|---|
| | RMSE ↓ | Pearson's r ↑ | RMSE ↓ | Pearson's r ↑ |
| GCN | 0.832 ± 0.001 | 0.527 ± 0.001 | 0.599 ± 0.006 | 0.809 ± 0.002 |
| DAGNN | 0.264 ± 0.004 | 0.964 ± 0.001 | 0.122 ± 0.004 | 0.991 ± 0.000 |
| D-VAE | 0.384 ± 0.002 | 0.920 ± 0.001 | 0.281 ± 0.004 | 0.964 ± 0.001 |
| S-VAE | 0.478 ± 0.002 | 0.873 ± 0.001 | 0.499 ± 0.006 | 0.873 ± 0.002 |
| GraphRNN | 0.726 ± 0.002 | 0.669 ± 0.001 | 0.779 ± 0.007 | 0.634 ± 0.001 |
| Graphormer | 0.352 ± 0.002 | 0.936 ± 0.001 | 0.181 ± 0.004 | 0.971 ± 0.001 |
| SAN | 0.311 ± 0.003 | 0.950 ± 0.001 | 0.158 ± 0.005 | 0.989 ± 0.001 |
| **UGC-GNN (GIN)** | **0.253 ± 0.002** | **0.964 ± 0.001** | **0.120 ± 0.004** | **0.992 ± 0.001** |

Table 4: Experimental results on DAG datasets. **Best results** are highlighted.

| Methods | Mayo | | RosMap | | Cancer | |
|---|---|---|---|---|---|---|
| | Accuracy ↑ | F1 score ↑ | Accuracy ↑ | F1 score ↑ | Accuracy ↑ | F1 score ↑ |
| SANEpool | 0.501 ± 0.041 | 0.486 ± 0.036 | 0.492 ± 0.036 | 0.507 ± 0.040 | 0.511 ± 0.032 | 0.508 ± 0.047 |
| Decagon | 0.561 ± 0.049 | 0.535 ± 0.021 | 0.520 ± 0.036 | 0.571 ± 0.032 | 0.593 ± 0.039 | 0.561 ± 0.042 |
| DimiG | 0.561 ± 0.049 | 0.535 ± 0.021 | 0.520 ± 0.036 | 0.571 ± 0.032 | 0.593 ± 0.039 | 0.561 ± 0.042 |
| TransSynergy | 0.532 ± 0.019 | 0.547 ± 0.028 | 0.540 ± 0.025 | 0.558 ± 0.032 | 0.588 ± 0.029 | 0.544 ± 0.035 |
| MLA-GNN | 0.589 ± 0.033 | 0.614 ± 0.036 | 0.595 ± 0.028 | 0.622 ± 0.040 | 0.631 ± 0.035 | 0.640 ± 0.037 |
| **UGC-GNN (GIN)** | **0.624± 0.036** | **0.713 ± 0.022** | **0.701± 0.025** | **0.689± 0.019** | **0.714± 0.011** | **0.701± 0.032** |

Table 5: Comparison to powerful DL baselines in Bioinformatics. Shown is the mean ± s.d. of 5 runs with different random seeds. **Best results** are highlighted.

help to find the universal graph canonization. Table 3 and Table 4 show that UGC-GNN consistently achieves the state-of-the-art (SOTA) performance and can provide a performance improvement up to 31% over GNN baselines.

**Efficiency.** Another significant advantage of enhancing GNNs with universal graph canonization is the time/space complexity. Suppose the graph has $n$ nodes with a maximum degree $d$, then UGC-GNN has a space complexity of $\mathcal{O}(n)$ and each layer takes $\mathcal{O}(nd)$ operations. In contrast, high-order GNNs like PPGN has a space complexity of $\mathcal{O}(n^2)$ for each layer and a $\mathcal{O}(n^3)$ time complexity; Subgraph-based GNNs (to radius $r$) has a $\mathcal{O}(nd^r)$ space complexity and each layer takes $\mathcal{O}(nd^{r+1})$ operations. In many real-world tasks especially in bioinformatics, graphs has a large $n$ and $r$, which makes existing dominant expresive GNNs (high-order GNNs and subgraph-based GNNs) unsuitable. Table 3 indicates that existing dominant expresive GNNs can easily have the our-of-memory (OOM) problem (like ESAN and DropGNN), while Appendix I shows that UGC-GNN can significantly reduce the computation time. Sepcifically, UGC-GNN only requires about $\frac{1}{4}$ and $\frac{1}{10}$ average time per epoch compared to subgraph-based GNN (NGNN) and high-order GNN (PPGN), respectively.

**The optimal solution to the challenging gene network analysis. Table 5.** Gene networks are the prevailing data structure in Bioinformatics (Podolsky & Greene, 2011; Carew et al., 2008). Recent years, it has emerged as a popular topic of developing the de facto standard for effective representation learning methods over gene networks, which leads to potential advancements in clinical applications including drug synergy prediction (Hopkins, 2008; Podolsky & Greene, 2011), Alzheimer's disease (AD) detection (Song et al., 2019; Qin et al., 2022) and cancer subtype classification (Lu & Han, 2003). In the experiment, we compare UGC-GNN against existing SOTA deep learning (DL) models for gene network analysis. Table 5 indicates that UGC-GNN outperforms these DL models and is the optimal solution to the challenging gene network analysis.

## 5 CONCLUSION AND DISCUSSIONS

In this paper, we propose to enhance GNNs through graph canonization that generates discrete colouring of nodes as their positional encodings to introduce node asymmetry. We theoretically show that graph-canonization-enhanced GNNs (i.e. GC-GNNs) maximize the expressivity in distinguishing non-isomorphic graphs at the cost of model stability, and then design a universal graph canonization to ameliorate this trade-off under a widely applicable sufficient condition that is satisfied in numerous real-world graph datasets. Massive experiments demonstrate the capability and great potential of GC-GNN in general graph representation learning; Furthermore, GNNs enhanced by the universal graph canonization (i.e. UGC-GNNs) show superior predictive ability and consistently achieve the state-of-the-art performance when the sufficient condition is satisfied, providing the optimal solution to many real-world applications like the challenging gene network representation learning in bioinformatics.

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

## A  REPRESENTATION LEARNING TASKS ON GENE NETWORKS AND OTHER GRAPH BENCHMARKS

In the natural world, genes don't operate independently and always function as part of a set of genes. Hence, they can be regulated as the graph modality based on reported genetic interactions that underlie phenotypes in a variety of bioinformatical systems. Then, various computational tasks on gene networks (graphs) are proposed in recent bioinformatics to numerically analyze contribution of genes to complex disease in humans. In this work, we take two long-standing challenging tasks in bioinformatics, Alzheimer's disease (AD) classification and cancer subtype classification, as example, and introduces corresponding gene-network datasets. Here we compare the these gene network datasets against popular graph benchmark datasets to illustrate challenges of graph representation learning over gene networks.

| Dataset | Ave. # nodes | Ave. # edges | # Tasks | Task Type | Metric |
|---------|-------------|-------------|---------|-----------|--------|
| Mayo | 3000 | 60000 | 2 | Classification | Accuracy & F1 |
| RosMap | 3000 | 60000 | 2 | Classification | Accuracy & F1 |
| Cancer Subtype | 3800 | 48000 | 7 | Classification | Accuracy & F1 |
| NA | 8 | 12 | 1 | Regression | RMSE & Pearson'r |
| BN | 10 | 15 | 1 | Regression | RMSE & Pearson'r |
| ogbg-molhiv | 26 | 55 | 2 | Classification | ROC-AUC |
| ZINC | 23 | 50 | 1 | Regression | MAE |
| D&D | 284 | 1431 | 2 | Classification | Accuracy |
| MUTAG | 18 | 39 | 2 | Classification | Accuracy |
| PROTEINS | 39 | 146 | 2 | Classification | Accuracy |
| PTC-MR | 14 | 29 | 2 | Classification | Accuracy |
| ENZYMES | 33 | 124 | 6 | Classification | Accuracy |

Table 6: Comparison of gene networks and graphs in popular benchmark datasets

Table 6 presents preliminary results. Compared to graphs in popular benchmarks, gene networks always contain significantly large number of nodes (denoted as $n$) as well as the number of edges (denoted as $m$), which limits the applicability of popular expressive GNNs due to the complexity consideration. For subgraph-based GNNs like NGNN (Zhang & Li, 2021), the space complexity grows exponentially as the average node degree $\frac{m}{n}$ increases. For high-order GNNs, such as k-WL (Morris et al., 2019) and k-FWL (Grohe, 2021), that mimic the high-order WL algorithms, the stable colourings/representations can be computed in $\mathcal{O}(k^2 n^{k+1} \log n)$ with a space complexity of $\mathcal{O}(n^k)$. Consequently, it is hard to apply these methods to gene network representation learning.

## B  MORE DISCUSSION OF GC-GNN AND UGC-GNN

In this section, we provide more discussion of GC-GNN.

- **Part 1:** Limitations of GC-GNN and UGC-GNN.
- **Part 2:** The complexity of GC-GNN and UGC-GNN.

**Part 1 (Table 7)**  One significant limitation of GC-GNN and UGC-GNN is that the (universal) graph canonization fail to consider the heterogeneity among edges. In addition, GC-GNN is also not guaranteed to be suitable for all graph learning tasks. Here we empirically test GC-GNN on moleculr datasets, ogbg-molhiv and ogbg-molpcba, in Open Graph Benchmark (Hu et al., 2020). Table 7 provides empirical results. As we can see, GC-GNN still improve the performance over backbone GNNs and GC-GNN can achieve competitive performance to the powerful GNN baseline: ESAN with ED policy. However, we also notice that the improvement of GC-GNN is not significant as TU datasets and GC-GNN can not achieve the SOTA performance. In order to make fair comparison, we set the number of epochs for training to be a hyper-parameter.

**Part 2**  The main concern of complexity comes from the graph canonization algorithms. As we discussed in the main paper (section 1.1, other related works), although graph canonization is a well-know NP-complete problem, practical canonization tools such as Nauty (McKay & Piperno, 2014) and Bliss (Junttila & Kaski, 2012), can effectively solve the problem in practice with an average time

|  | ogbg-molhiv ( AUC ↑) | ogbg-molpcba (AP ↑ ) |
|---|---|---|
| GCN (backbone) | $0.7501 \pm 0.0140$ | $0.2422 \pm 0.0034$ |
| ESAN (GCN + ED) | $0.7559 \pm 0.0120$ | $0.2521 \pm 0.0040$ |
| **GN-GCN** | $0.7552 \pm 0.0156$ | $0.2510 \pm 0.0047$ |
| GIN (backbone) | $0.7744 \pm 0.0098$ | $0.2703 \pm 0.0023$ |
| ESAN (GIN + ED) | $0.7803 \pm 0.0170$ | $0.2782 \pm 0.0036$ |
| **GN-GIN** | $0.7785 \pm 0.0195$ | $0.2761 \pm 0.0043$ |

Table 7: Empirical results on molecular datasets.

complexity of $\mathcal{O}(n)$. The process of computing the canonical forms of graphs can be implemented in the graph pre-process phase and we only need to perform practical graph canonization tools once for each graph. Furthermore, as the output discrete colouring $\{\rho(v_1|G), \rho(v_2|G)...\rho(v_n|G)\}$ of graph canonization are used as positional encodings of nodes, the additional space complexity is also $\mathcal{O}(n)$. Hence, compared to dominant expressive GNNs like subgraph-based GNNs and high-order GNNs, GC-GNN (UGC-GNN) is much more efficient with a significant low space and computation cost. For instance, on ogbg-molhiv, GC-GNN (GIN) takes 39 seconds per epoch, while Nested GIN takes 168 seconds.

## C  PROOF OF LEMMA 2.3 AND LEMMA 2.4

We first prove Lemma 2.3. Since discrete colouring $\{\rho(v_1|G), \rho(v_2|G), ..., \rho(v_n|G)\}$ generated by graph canonization provides a unique way to label nodes for graphs in the same isomorphic group $ISO(G)$, then the isomorphism of two graphs can be determined by checking whether the node pairs of the same labels share the same edge relation. When the graph convolution layers on GNN $g$ are injective, it can injectively map the pair of a node and the set of its' neighboring nodes. then if two graphs get different sets of learnt node embeddings by the GNN $g$, there must exists at least one pair of node label and the set of its' neighboring nodes' labels are different. Then, these two graphs are not isomorphic. On the other hand, when two graphs are not isomorphic, there must be at least one pair of nodes with the same labels, while the pair of nodes are connected in one graph yet disconnected in another graph. Then the graph convolution layer will output different embeddings for each node in the node pair.

Next, we prove Lemma 2.4 to support theoretical results about the stability of GNNs and GC-GNNs. Before proving our lemma, we first introduce some necessary preliminaries that we will later use in the proofs.

**Preliminary 1: Decomposition of message passing layer of GNNs**  The message passing scheme adopted in popular GNNs iteratively updates a node's representation/embedding according to the multiset of its neighbors' representations/embeddings and its current representation/embedding. Let $h_v^t$ denotes the representation of $v$ in layer $t$, the massage passing scheme is given by:

$$h_v^{t+1} = \mathcal{M}(h_v^t, \{h_u^t|(u,v) \in E\})$$
$$= \mathcal{U}(h_v^t, \mathcal{A}(\{h_u^t|(u,v) \in E\}))$$

Here, $\mathcal{A}$ is an aggregation function on the multiset $S = \{h_u^t|(u,v) \in E\})$ and $\mathcal{U}$ is an update function.

**Definition C.1** *A function $f$ is claimed as a L-stable multiset function if 1) $\sum_{h \in S} f(h)$ is unique for each multiset $S$ of bounded size; and 2) for any two multisets $S_1$ and $S_2$, $||\sum_{h \in S_1} f(h) - \sum_{h' \in S_2} f(h')|| \leq L \times d_S(S_1, S_2)$, where $d_S(S_1, S_2) = |S_1| + |S_2| - |S_1 \cap S_2|$.*

The L-stable multiset function $f$ provides a unique mapping between the multiset space and representation space, while distance between multisets in the representation space are bounded through the constant multiplier L.

**Corollary C.2** *Assume that input feature space $\mathbb{H}$ is countable. Any message passing function $\mathcal{M}$ over pairs $(h, S)$, where $h \in \mathbb{H}$ and $S \subset \mathbb{H}$, can be decomposed as $\mathcal{M}(h, S) = \phi((1 + \epsilon)f(h) +$*

$\sum_{h' \in S} f(h'))$ *for some L-stable multiset function* $f$, *some function* $\phi$ *and infinitely many choices of* $\epsilon$.

This corollary C.2 can be obtained by following the Lemma 5 and Corollary 6 in Xu et al. (2019). Basically, since the space $\mathbb{H}$ is countable, we can always get a mapping $Z : \mathbb{H} \to \mathbb{N}$ from $h \in \mathbb{H}$ to natural numbers. As $S$ are bounded multiset, there always exists an upper bound $B$ of their cardinality such that $|S| < B$ for any $S$. Hence, the example function $f(h) = B^{-Z(h)}$ satisfies both the uniqueness condition as well as the inequality condition in Definition C.1 where the constant $L = \frac{1}{B}$.

**Preliminary 2: Graph canonization and individualization-and-refinement paradigm**   Here we provide an extended discussion on the graph canonization and individualization-refinement paradigm. To address the complex graph isomorphism/ graph canonization problem, practical graph canonization tools resort to the individualization-and-refinement paradigm, where the color refinement and individualization steps are iteratively performed to get a discrete colouring.

- 1. Color refinement step: The color refinement algorithm aims to recolor nodes in a graph by similarity. The algorithm starts with some initial node colors, then the algorithm updates node's color round by round, and in each round, two nodes with the same color will get different new color if the multiset of neighboring colors are different. This process continues until an equitable colouring is obtained such that the node colors will not change even if another color refinement round happens.
- 2. Individualization step: When a stable colouring generated by the color refinement step is discrete, then returns a order of nodes in the canonical form. However, in many cases, the stable colouring are not discrete. Then the individualization step selects a node in a color class with more than one node and assigns a new (unseen) color to the node. Then the color refinement step is implemented again.

Specifically, in the color refinement step, the new colors are obtained by lexicographically sorting the pair of node current color and it's multiset of neighboring colors. Hence, the new order of new node colors always follows that of the previous colors. That is, at round $t$, if two nodes $v$, $u$ have different colors $c^t(u)$ and $c^t(v)$ such that $c^t(u) < c^t(v)$, then after one round of color refinement, the new colors of these two nodes have $c^{t+1}(u) < c^{t+1}(v)$.

Above individualization-refinement paradigm in practical graph canonization tools provide a solution to obtain a discrete coloring, yet not in a canonical way. That is, generated discrete coloring is not guaranteed to be the same for graphs in the same isomorphic class. To fix this, practical tools usually branch on all nodes of the same color in the individualization step and individualizes one node in each branch. Then a tree of colouring can be obtained such that each leaf of the tree is a discrete coloring of the input grpah $G$. Then the final discrete colouring is selected, for example, as the leaf with the lexicographically minimal string that consists of the rows of the adjacency matrix according to the discrete colouring (i.e. node orders). More details can be found in McKay & Piperno (2014).

***Proof.* (Part one: A GNN model** $g$ **is stable under** $X$**)**   Here, we assume the function $\phi$ in the decomposition of message passing layer of GNNs is K-Lipschitz. A function $\phi : (X, d_x) \to (Y, d_y)$ between two metric spaces is K-Lipschitz if $d_y(\phi(x_1), \phi(x_2)) \leq K d_x(x_1, x_2)$ for any $x_1, x_2 \in X$, where K is a constant.

**Corollary C.3** *MLPs are K-Lipschitz.*

In popular message passing GNNs, $\phi$ is always modeled by a MLP. Hence, we need to prove corollary C.3 to use the K-Lipschitz assumption. For each MLP layer that characterized by the trainable parameter tensors $W_i$ and $b_i$ (bias), it can be represented as $\sigma(W_i x + b_i)$, where $\sigma$ is the activation function. Then we have $||\sigma(W_i x_1 + b_i) - \sigma(W_i x_2 + b_i)|| = ||\frac{\partial \sigma}{\partial x} W_i (x_1 - x_2)||$. Since the activation function $\sigma$ usually takes ReLU/sigmoid/tanh, it's straightforward that $||\frac{\partial \sigma}{\partial x}||$ is bounded by a constant $K_1$. Then we get $||\frac{\partial \sigma}{\partial x} W_i (x_1 - x_2)|| \leq K_1 ||W_i||_2 ||x_1 - x_2||$, where $K_1 ||W_i||_2$ is a constant independent of $x_1$ and $x_2$. Hence, we show that MLPs are K-Lipschitz.

Next, let's prove the theoretical result. Without loss of generality, we assume that the GNN contains a single message passing layer. For any two graphs $G^{(1)} = (A^{(1)}, X^{(1)})$ and $G^{(2)} = (A^{(2)}, X^{(2)})$,

let $\pi^* \in \Pi(n)$ denotes the optimal permutation operation that best aligns the $G_1$ and $G_2$, and $P^*$ is the corresponding permutation matrix. Then we have,

$$||g(A^{(1)}, X^{(1)}) - g(A^{(2)}, X^{(2)})||$$

$$=|| \sum_{v \in G^{(1)}} \phi((1+\epsilon)f(X_v^{(1)}) + \sum_{v' \in \mathcal{N}(v|G^{(1)})} f(X_{v'}^{(1)})) - \sum_{u \in G^{(2)}} \phi((1+\epsilon)f(X_u^{(2)}) + \sum_{u' \in \mathcal{N}(u|G^{(2)})} f(X_{u'}^{(2)}))|| \quad (1)$$

$$\leq \sum_{v \in G^{(1)}} ||\phi((1+\epsilon)f(X_v^{(1)}) + \sum_{v' \in \mathcal{N}(v|G^{(1)})} f(X_{v'}^{(1)})) - \phi((1+\epsilon)f(X_{\pi^*(v)}^{(2)}) + \sum_{u' \in \mathcal{N}(\pi^*(v)|G^{(2)})} f(X_{u'}^{(2)}))|| \quad (2)$$

$$\leq K \sum_{v \in G^{(1)}} ||(1+\epsilon)f(X_v^{(1)}) + \sum_{v' \in \mathcal{N}(v|G^{(1)})} f(X_{v'}^{(1)}) - (1+\epsilon)f(X_{\pi^*(v)}^{(2)}) - \sum_{u' \in \mathcal{N}(\pi^*(v)|G^{(2)})} f(X_{u'}^{(2)})|| \quad (3)$$

$$\leq K(1+\epsilon) \sum_{v \in G^{(1)}} ||f(X_v^{(1)}) - f(X_{\pi^*(v)}^{(2)})|| + K \sum_{v \in G^{(1)}} || \sum_{v' \in \mathcal{N}(v|G^{(1)})} f(X_{v'}^{(1)}) - \sum_{u' \in \mathcal{N}(\pi^*(v)|G^{(2)})} f(X_{u'}^{(2)})|| \quad (4)$$

$$\leq K \times L \times (1+\epsilon)(\sum_v X_v^{(1)} \neq X_{\pi^*(v)}^{(2)}) + K \times L \times d_S(\mathcal{N}(v|G^{(1)}), u' \in \mathcal{N}(\pi^*(v)|G^{(2)})) \quad (5)$$

$$\leq K \times L \times (1+\epsilon)d(G^{(1)}, G^{(2)}) \quad (6)$$

Here, we use the K-Lipschitz property in the step (4), and use the property of L-stable multiset function in step (6). Furthermore, $\mathcal{N}(v|G^{(1)})$ denotes the set of neighboring nodes of $v$ in $G^{(1)}$, and can be characterized by the $v$th row of adjacency matrix $A^{(1)}$. $\mathcal{N}(\pi^*(v)|G^{(2)})$ denotes the set of neighboring nodes of $v$'s image in $G^2$, and can be characterized by the $v$th row of adjacency matrix $P^* A^{(2)} P^{*T}$. Hence, we show that GNN $g$ is stable under $X$ with a constant $C$ of $K \times L \times (1+\epsilon)$.

***Proof.*** **(Part two: A GNN model $g$ is not stable under $X \oplus P$)**  Here we provide a counter example. Let $G^{(1)}$ be a graph of $n$ nodes such that 1) there is no node symmetry in the graph; 2) the node $v_n$ has an initial color (integer feature) that is strictly larger than the colors of other nodes. Then, we obtain a graph $G^{(2)}$ by changing the initial color of node $v_n$ in $G^{(1)}$ to another color which is strictly smaller than the colors of other nodes. Then, we know that there is no node symmetry in the graph $G^{(2)}$, either. Furthermore, it's straightforward that the best graph matching $\pi^* \in \Pi(n)$ between $G^{(1)}$ and $G^{(2)}$ is $\pi(i) = i$ for $\forall i = 1, 2, ...n$ and the corresponding permutation matrix $P^* = I$ is an identity matrix.

Since there is no node symmetry in $G^{(1)}$ and $G^{(2)}$, the color refinement step in graph normalization tools will generate discrete colourings for $G^{(1)}$ and $G^{(2)}$. Let $\{\rho(v_1|G^{(1)}), \rho(v_2|G^{(1)}), ..., \rho(v_n|G^{(1)})\}$ and $\{\rho(v_1|G^{(2)}), \rho(v_2|G^{(2)}), ..., \rho(v_n|G^{(2)}\}$ denote the corresponding discrete colourings. As we discussed in preliminary 2, the output discrete colouring from color refinement will keep the order of initial colors. Hence, we know that $\rho(v_n|G^{(1)}) = n$, $\rho(v_n|G^{(2)}) = 1$ and $\rho(v_i|G^{(1)}) = \rho(v_i|G^{(2)}) + 1$ for $i = 1, 2, ...n - 1$. Thus, we get $\rho(v_i|G^{(1)}) \neq \rho(v_i|G^{(2)})$ for $i = 1, 2, ..., n$, indicating that $P_v^{(1)} \neq P_v^{(2)}$ for $\forall v$. Thus, we have,

$$||g(A^{(1)}, X^{(1)}) - g(A^{(2)}, X^{(2)})||$$

$$\geq || \sum_{v \in G^{(1)}} (1+\epsilon)(f(X_v^{(1)} + P_v^{(1)}) - f(X_v^{(2)} + P_v^{(2)}))|| \quad (7)$$

$$\geq (1+\epsilon) \times |G^{(1)}| \times \frac{1}{B} \quad (8)$$

$$\geq 1 = d(G^{(1)}, G^{(2)}) \quad (9)$$

Since the function $f$ is defined on the multiset whose cardinality is bounded by the overall graph size $n$, we get $B \leq |G^{(1)}|$.

## D  PROOF OF THEOREM 3.2

***Proof.*** Following the proof of Lemma 2.4, we consider the same decomposition scheme $\mathcal{M}(h, S) = \phi((1+\epsilon)f(h) + \sum_{h' \in S} f(h'))$ of the message passing layer, yet the input space is $X \oplus P$ (where $P$

is the 2-dimensional tensor of one-hot encodings of discrete colouring generated by universal graph normalization), instead of the input feature $X$.

Then, let's consider any pairs $\mathcal{N}(v|G^{(1)}))$ and $\mathcal{N}(\pi^*(v)|G^{(2)}))$. Since the discrete colouring in any common subgraph $G^{'}$ of $G^{(1)}$ and $G^{(2)}$ are identical, the number of same items in multisets $\{X_{v^{'}}^{(1)}|v^{'} \in \mathcal{N}(v|G^{(1)})\}$ and $\{X_{u^{'}}^{(2)}|u^{'} \in \mathcal{N}(\pi^*(v)|G^{(2)})\}$ will not change if we replace $X_{v^{'}}^{(1)}$ with $X_{v^{'}}^{(1)} + P_{v^{'}}^{(1)}$, and $X_{u^{'}}^{(2)}$ with $X_{u^{'}}^{(2)} + P_{u^{'}}^{(2)}$. Then we get,

$$\sum_{v^{'} \in \mathcal{N}(v|G^{(1)})} f(X_{v^{'}}^{(1)} + P_{v^{'}}^{(1)}) - \sum_{u^{'} \in \mathcal{N}(\pi^*(v)|G^{(2)})} f(X_{u^{'}}^{(2)} + P_{u^{'}}^{(2)})$$
$$= \sum_{v^{'} \in \mathcal{N}(v|G^{(1)})} f(X_{v^{'}}^{(1)}) - \sum_{u^{'} \in \mathcal{N}(\pi^*(v)|G^{(2)})} f(X_{u^{'}}^{(2)}) \tag{10}$$

Thus, we have

$$||g(A^{(1)}, X^{(1)}) - g(A^{(2)}, X^{(2)})||$$
$$= || \sum_{v \in G^{(1)}} \phi((1+\epsilon)f(X_v^{(1)} + P_v^{(1)}) + \sum_{v^{'} \in \mathcal{N}(v|G^{(1)})} f(X_{v^{'}}^{(1)} + P_{v^{'}}^{(1)}))$$
$$- \sum_{u \in G^{(2)}} \phi((1+\epsilon)f(X_u^{(2)} + P_u^{(2)}) + \sum_{u^{'} \in \mathcal{N}(u|G^{(2)})} f(X_{u^{'}}^{(2)} + P_{u^{'}}^{(2)}))|| \tag{11}$$
$$\leq K \sum_{v \in G^{(1)}} ||(1+\epsilon)f(X_v^{(1)} + P_v^{(1)}) + \sum_{v^{'} \in \mathcal{N}(v|G^{(1)})} f(X_{v^{'}}^{(1)} + P_{v^{'}}^{(1)})$$
$$- (1+\epsilon)f(X_{\pi^*(v)}^{(2)} + P_{\pi^*(v)}^{(2)}) - \sum_{u^{'} \in \mathcal{N}(\pi^*(v)|G^{(2)})} f(X_{u^{'}}^{(2)} + P_{u^{'}}^{(2)})|| \tag{12}$$
$$\leq K(1+\epsilon) \sum_{v \in G^{(1)}} ||f(X_v^{(1)} + P_v^{(1)}) - f(X_{\pi^*(v)}^{(2)} + P_{\pi^*(v)}^{(2)})||$$
$$+ K \sum_{v \in G^{(1)}} || \sum_{v^{'} \in \mathcal{N}(v|G^{(1)})} f(X_{v^{'}}^{(1)} + P_{v^{'}}^{(1)}) - \sum_{u^{'} \in \mathcal{N}(\pi^*(v)|G^{(2)})} f(X_{u^{'}}^{(2)} + P_{u^{'}}^{(2)})|| \tag{13}$$
$$= K(1+\epsilon) \sum_{v \in G^{(1)}} ||f(X_v^{(1)} + P_v^{(1)}) - f(X_{\pi^*(v)}^{(2)} + P_{\pi^*(v)}^{(2)})||$$
$$+ K \sum_{v \in G^{(1)}} || \sum_{v^{'} \in \mathcal{N}(v|G^{(1)})} f(X_{v^{'}}^{(1)}) - \sum_{u^{'} \in \mathcal{N}(\pi^*(v)|G^{(2)})} f(X_{u^{'}}^{(2)})|| \tag{14}$$
$$\leq K \times L \times (1+\epsilon)d(G^{(1)}, G^{(2)}) \tag{15}$$

where we use the equation (11) in the step (14) to get step (15).

## E  PROOF OF LEMMA 3.3

*Proof.* Since the function $l(v|\mathbb{G}) : V \rightarrow \mathbb{N}$ is an injective function, it can distinguish nodes in each graph according to $l(v|\mathbb{G})$. Furthermore, since for $\forall\, v_1, u_1 \in G_1, v_2, u_2 \in G_2, G_1, G_2 \in \mathbb{G}$ such that $l(v_1|\mathbb{G}) = l(v_2|\mathbb{G})$ and $l(u_1|\mathbb{G}) = l(u_2|\mathbb{G})$, we have $(v_1, u_i) \in E_1 \leftrightarrow (v_2, u_2) \in E_2$, we know that the connectivity between node pairs $(v, u)$ of the same label pairs $(l(v|\mathbb{G}), l(u|\mathbb{G}))$ is shared among all graphs. Let N be the total number of all potential different labels $l(v|\mathcal{G})$, then we can expand each graph to a larger graph of size N, where each node $v$ has an order/position of $l(v|\mathcal{G})$, and rest positions are padded by dummy nodes that is not connected with any other nodes. Then, any common subgraph $G^{'}$ of $G_1$ and $G_2$, is also a common subgraph of their converted larger graphs, and it is obivous that the orders of nodes in each common subgraph of these generated larger graphs are identical.

## F    Proof of Lemma 3.4

*Proof.* Let $G = (A, X)$ be an arbitrary graph of size $n$, and $\forall P \in \Pi(n)$ where $\Pi(n)$ is the permutation group. $\{l(v_1|\mathbb{G}), l(v_2|\mathbb{G})...l(v_n|\mathbb{G})\}$ is the discrete colouring of $G$ generated by the universal graph canonization. The discrete colouring is invariant to the permutation operation $P$ as graphs from the same isomorphic class has the same canonical form (so as the same output discrete colouring from universal graph canonization). Here, we use $L$ to denote a 2-dimension tensor of one-hot features of this discrete colouring. Then, let $G' = (PAP^T, PX)$ ($G'$ is isomorphic to $G$), then, the corresponding discrete colouring tensor is $PL$. Let $\mathcal{M}$ be a stack of message passing layers, then we have $P\mathcal{M}(A, X) = \mathcal{M}(PAP^T, PX)$ for $\forall X, A$. Let $\mathcal{R}$ denote the readout function of UNGNN, that is $\mathcal{R}(\{h_v|v \in V\}) = \sum_{v \in V} W_{l(v|\mathbb{G})} h_v$, then $\mathcal{R}$ is invariant to the order of $\{h_v|v \in V\}$ as the corresponding weight matrix $W_{l(v|\mathbb{G})}$ is solely decided by node's discrete color $l(v|\mathbb{G})$. Hence, UNGNN $g$ is a composition of functions $\mathcal{R}$ and $\mathcal{M}$, then we get:

$$
\begin{aligned}
g(PAP^T, PX) &= \mathcal{R}(\mathcal{M}(PAP^T, PX + PL)) \\
&= \mathcal{R}(\mathcal{M}(PAP^T, P(X + L))) \\
&= \mathcal{R}(P\mathcal{M}(A, X + L)) \\
&= \mathcal{R}(\mathcal{M}(A, X + L)) = g(A, X)
\end{aligned}
$$

## G    More Ablation Results

| Methods | Mayo | | RosMap | | Cancer Subtype | |
|---|---|---|---|---|---|---|
| | Accuracy ↑ | F1 score ↑ | Accuracy ↑ | F1 score ↑ | Accuracy ↑ | F1 score ↑ |
| GIN | $0.496 \pm 0.042$ | $0.484 \pm 0.036$ | $0.471 \pm 0.039$ | $0482 \pm 0.041$ | $0.537 \pm 0.045$ | $0.512 \pm 0.047$ |
| GCN | $0.561 \pm 0.049$ | $0.535 \pm 0.021$ | $0.520 \pm 0.036$ | $0.571 \pm 0.032$ | $0.593 \pm 0.039$ | $0.561 \pm 0.042$ |
| GIN + GC | $0.483 \pm 0.026$ | $0.472 \pm 0.031$ | $0.486 \pm 0.041$ | $0.510 \pm 0.037$ | $0.539 \pm 0.041$ | $0.532 \pm 0.069$ |
| GCN + GC | $0.522 \pm 0.019$ | $0.539 \pm 0.031$ | $0.508 \pm 0.032$ | $0.527 \pm 0.031$ | $0.544 \pm 0.025$ | $0.582 \pm 0.042$ |
| GIN + UGC | $0.561 \pm 0.027$ | $0.570 \pm 0.031$ | $0.697 \pm 0.041$ | $0.624 \pm 0.037$ | $0.589 \pm 0.041$ | $0.562 \pm 0.069$ |
| GCN + UGC | $0.572 \pm 0.021$ | $0.619 \pm 0.037$ | $0.658 \pm 0.030$ | $0.621 \pm 0.021$ | $0.619 \pm 0.025$ | $0.581 \pm 0.031$ |
| **UGC-GNN (GIN)** | $\mathbf{0.624 \pm 0.036}$ | $\mathbf{0.713 \pm 0.022}$ | $\mathbf{0.701 \pm 0.025}$ | $\mathbf{0.689 \pm 0.019}$ | $\mathbf{0.714 \pm 0.011}$ | $\mathbf{0.701 \pm 0.032}$ |
| **UGC-GNN (GCN)** | $\mathbf{0.603 \pm 0.031}$ | $\mathbf{0.652 \pm 0.020}$ | $\mathbf{0.724 \pm 0.021}$ | $\mathbf{0.697 \pm 0.019}$ | $\mathbf{0.691 \pm 0.013}$ | $\mathbf{0.706 \pm 0.029}$ |

Table 8: Ablation study results. GC = graph canonization, UGC = universal graph canonization. **Best results** are highlighted.

**Ablation study reults. Table 8.** This experiment tests the effectiveness of different components in UGC-GNN and empirically supports theoretical findings in the paper. When comparing GIN (GCN), GIN + GC (GCN + GC), and GIN + UGC (GCN + UGC), (1) we find GNNs with general graph canonization usually causes the decrease of the testing performance (i.e. GIN + GC ¡ GIN, GCN + GC ¡ GCN). This finding aligns well with Lemma 2.4, indicating that stability of GNN is of great practical importance in real datasets especially when graphs are large. (2) We also observe that GNNs with universal graph canonization significantly enhance the model performance, and the observation empirically supports the Lemma 3.3. Furthermore, we also find that GIN (GCN)¡ GIN + UGC (GCN + UGC) ¡ and UGC-GNN (GIN or GCN). Consequently, the canonical information from the discrete colouring $\{l(v_1|\mathbb{G}), l(v_2|\mathbb{G})...l(v_n|\mathbb{G})\}$ enhance the base GNN architecture from the perspective of both message passing process and the readout function.

## H    Graph Structured Data in Bioinformatics and their GNN Baselines

The dominant graph-structured biological data in bioinformatics can be categorized into three types: (1) molecular graphs, (2) biological interaction graphs and (3) gene networks/graphs. Different types of these biological graph data have different SoTA GNN baselines.

In molecular graphs, atoms or other chemical compounds are used as nodes, and the bonds are formulated as the edges. Molecular graphs are homogeneous graphs and are widely used as benchmark

datasets in the field of representation learning on graphs and GNNs (graph neural networks). Typical molecular graph benchmark datasets include ZINC, QM9 Ramakrishnan et al. (2014), Molhiv in OGB (Open Graph Benchmark), etc. Dominant baseline deep learning models for these molecular graph datasets are popular GNN models, such as GIN, GCN, GINE, NGNN, PPGN, and these baselines are used in our experimnets.

In biological interaction graphs, nodes represent genes, drugs, disease, RNA, etc, and an edge indicates the existence of a known association between entities connected by the edge. Thus, biological interaction graphs are heterogeneous graphs. Typical biological interaction graphs include gene-drug interaction networks in the drug synergy prediction task, drug-protein interaction networks Zitnik et al. (2018) in side effect prediction tasks, and miRNA-disease interaction networks  Pan & Shen (2019) in the disease classification task. For the representation learning tasks on biological interaction graphs, various task-specific graph neural networks are proposed, and here we provide some popular examples: TransSynergy  Liu & Xie (2021) and SANEpool  Dong et al. (2023) for gene-drug interaction networks; Decagon for drug-protein interaction networks; and DimiG  Pan & Shen (2019) for miRNA-disease interaction networks. Overall, these task-specific GNNs focus more on how to effectively construct the biological interaction graphs and represent initial features related to genes/drugs/ disease/RNA. Instead of the architecture design of GNNs. Consequently, these task-specific GNNs may not be well generalized to other graph learning tasks such as the gene network representation learning. In the paper, we select TransSynergy, SANEpool, Decagon and DimiG as baselines, and compare them with the proposed UGC-GNN.

In gene graphs/networks, genes are used as nodes, and an edge is used to link two genes if there is a physical signaling interaction between the genes from the documented medical experiment. Similar to molecular graphs, gene graphs/networks are also homogeneous graphs. Very limited works are proposed for gene network representation learning. MLA-GNN is proposed for graph-level classification tasks on gene networks, thus it can be used as a good baseline in the field.

## I  COMPUTATION EFFICIENCY

In this experiment, we compare the of computation cost of GNNs enhanced by graph canonization techniques (UGC-GNN) against dominant expressive GNNs to emphasize the advantage in efficiency.(1) For high-order GNNs, PPGN has a space complexity of $\mathcal{O}(n^2)$ for each layer, while 1-2-3 GNNs has a space complexity of $\mathcal{O}(n^3)$ for each layer. Then, when we test 1-2-3 GNN on Mayo, RosMap and Cancer, 1-2-3 GNNs always meet the OOM (out-of-memory) problem . On the other hand, by reducing the embedding dimension and number of layers, PPGN can avoid OOM problem on these datasets as Table  3 in the main paper indicates; (2) For subgraph-based GNNs like NGNN (nested GNN), we also implement the similar strategy to restrict the radius of subgraphs to be smaller than 2, then NGNN can avoid the OOM problem, too. Consequently, here we compare the average computation time per epoch of UGC-GNN against NGNN and PPGN on large-scale datasets.

|  | Mayo | RosMap | Cancer |
|---|---|---|---|
| NGNN | 48s | 42s | 334s |
| PPGN | 143s | 126s | 1261s |
| **UGC-GNN (GIN)** | **13s** | **10s** | **89s** |

Table 9: Average computation time per epoch.

Table  9 demonstrates the experimental results. First, we find that UGC-GNN significantly reduces the computation complexity compared to high-order GNN PPGN, and this observation can be explained by the $\mathcal{O}(n^3)$ time complexity of PPGN and the large size of graphs. On the other hand, NGNN suffers from large subgraph sizes caused by the large average node degree in these datasets, then the extra computation burden is also larger than popular graph datasets with small average node degree (like OGB and TU datasets).

In addition, we need to point out that ESAN with EGO and EGO+ policy can be considered as a subgraph-based GNN. However, when the node-deleted subgraphs (ND) and the edge-deleted subgraphs (ED) polices are used, a graph is mapped to the set of subgraphs obtained by removing a single node or edge, then ESAN can be understood through the marking prism. In our experiments, ESAN is equipped with ED policy.

