# OpenReview forum: "Rethinking the Power of Graph Canonization in Graph Representation Learning with Stability"
_ICLR.cc/2024/Conference — ICLR 2024 poster_

### Official Review · Reviewer_RfLx · 2023-10-31

**Soundness:** 3 good
**Presentation:** 3 good
**Contribution:** 2 fair
**Rating:** 6
**Confidence:** 3

**Summary:**

This work focuses on the problems of the expressive power of graph neural networks (GNNs) where they aim to enhance the expressive power of GNNs by graph canonization. To this end, the paper first reveals the trade-off between the expressivity and stability in GNNs enhanced by graph canonization, showing that graph-canonization-enhanced GNNs maximize their expressive power at the cost of model stability. The paper further proposes a universal graph canonization to tackle this trade-off under a sufficient condition. The advantages of the proposed models against some GNN baselines are validated by their experimental results on both synthetic datasets and several benchmark datasets.

**Strengths:**

1. The paper is clear and well-structured.

 2. The theoretical results that reveal the trade-off between expressivity in distinguishing non-isomorphic graphs and the stability in GNNs enhanced by graph canonization are interesting. The theoretical findings have shown the limitations of applying graph canonization to enhance the expressive power of GNNs. The proposed universal graph canonization to tackle the trade-off between expressivity and stability is novel.

 3. The experimental results of the proposed model, UGC-GNN, on several bioinformatical graph datasets are impressive and consistently outperformed some well-known GNN baselines.

**Weaknesses:**

1. Lack of experiments to compare the expressive power of the proposed models with high-order GNNs [1, 2] and subgraph-based GNNs [3, 4] in distinguishing non-isomorphic graphs. Since high-order GNNs also enhance the expressive power of GNNs, it would be helpful to conduct experiments to compare the expressive power of the proposed models and high-order GNNs. Are graph-canonization-enhanced GNNs more powerful than high-order GNNs and k-WL test algorithms?

 2. Lack of computation cost comparison with k-WL-GNNs. High-order GNNs and subgraph-based GNNs may suffer from high computational cost when applied to large-scale graphs. The paper asserts that the proposed models are more efficient with a significantly low space and computation cost. It would be nice to provide empirical evidence to support the claim.

 3. As shown in Table 7, the GC-GNN doesn't consistently enhance the performance of the GNN backbone. In some instances, the performance even diminishes. It would be beneficial to delve deeper into which graph properties might influence this fluctuation in performance when integrating the GNN backbone with GC-GNN or UGC-GNN.

 4. The universal graph canonization problem is NP-hard. While the paper has provided a sufficient condition to compute the discrete coloring and discussed the applicability in several application scenarios, finding an injective function $l(v|\mathbb{G})$ might be elusive or challenging for general graph learning contexts. Furthermore, if there could be multiple choices of $l(v|\mathbb{G})$, it is unclear what the impact of applying different choices of $l(v|\mathbb{G})$ is on the performance of the proposed model.

Overall, the paper is well-written and easy to follow. The theoretical findings on the trade-off between expressive power and stability in graph-canonization-enhanced GNNs are interesting and the proposed universal graph canonization to tackle the trade-off is novel. Despite the potential high computational cost induced by the graph canonization algorithms and the effectiveness of the proposed models that might not be generalized to general graph datasets, the paper has provided some insights into the expressive power of GNNs enhanced by graph canonization methods. Therefore, the reviewer is inclined to accept the paper.

Reference

 [1] Weisfeiler and leman go neural: Higher-order graph neural networks, AAAI 2019.

 [2] Provably powerful graph networks, NeurIPS 2019.

 [3] Nested graph neural networks, NeurIPS 2021.

 [4] Equivariant Subgraph Aggregation Networks, ICLR 2022.

**Questions:**

Please refer to weaknesses

---

> ### Author Response · Authors · 2023-11-20
> **Author Response 1/2**
>
> We appreciate the reviewer’s constructive feedback and positive review.
>
> $\textbf{Q1:}$  Lack of experiments to compare the expressive power of the proposed models with high-order GNNs [1, 2] and subgraph-based GNNs [3, 4] in distinguishing non-isomorphic graphs.
>
> $\textbf{Re1:}$ We agree that adding more expressive GNNs as baselines can help to evaluate the significance of our method. In our paper, the suggested nested GNN is already included (i.e. NGNN), while PPGN is included in bioinformatical datasets (Table 3).  Before we present the supplementary experimental results, we want to clarify that the term ‘subgraph-based GNNs’ in our paper indicates GNNs that improve the expressivity by encoding induced k-hop neighborhood of nodes. Then subgraph-based GNNs model hierarchies of local isomorphism on neighborhood subgraphs.  Thus, ESAN [4] with node-deleted policy (ND) and edge-deleted policy (ED) can not be catogorized into this group. ESAN can be categorized as GNNs with markings [6] that slightly perturbs the input graphs for multiple times to execute GNNs and then aggregates the learnt final embeddings of these perturbed graphs in each run. Another well-known example of GNNs with markings is DropGNN [5], and more details of GNNs with markings are available in [6].
>
> Thus, in the supplementary experiments, we compare our methods against 1-2-3 GNN [1], PPGN [2] (on TU datasets), ESAN (ED) [4]  and DropGNN [5].
>
> ## Table 1. Supplementary experiments on TU datasets
> |Dataset| DD | MUTAG | PROTEINS | PTC-MR | ENZYMES |
> | :------:| :------: | :------: | :------: | :------: | :------: |
> | 1-2-3 GNN | 76.3 | 86.1 |75.5 | 60.9 | 32.7 |
> | PPGN| 78.5+- 5.1 | 90.6 +- 8.7 | $\textbf{77.2}$+- $\textbf{4.7}$ | 66.3 +- 8.6| 36.8 +- 5.9 |
> | ESAN | 79.7 +- 3.8| $\textbf{91.0}$ +- $\textbf{4.8}$ |75.8 +- 4.5| 65.7 +- 7.1 | 37.9 +- 6.3 |
> | DropGNN| 78.5+- 6.0| 90.4 +- 7.0 |76.3 +- 6.1 | 62.7 +- 8.4 | $\textbf{38.3}$ +-$\textbf{7.1}$ |
> | GC-GNN (ours) |$\textbf{91.3}$ +- $\textbf{9.7}$|86.2 +- 9.9 | 76.7 +- 5.1 | $\textbf{66.9}$+- $\textbf{7.1}$ |37.7 +- 6.9 |
>
>
> ## Table 2.  Supplementary experimental results on bioinformatical datasets
> |Dataset| Mayo | Mayo | RosMap | RosMap | Cancer | Cancer |  fMRI-ABIDE | fMRI-ABIDE |
> | :------:| :------: | :------: | :------: | :------: | :------: | :------: |  :------: | :------: |
> |Metric | Accuracy |F1 score| Accuracy |F1 score | Accuracy |F1 score | Accuracy |F1 score |
> | 1-2-3 GNN | OOM | OOM | OOM | OOM | OOM | OOM | 0.631 +- 0.041 | 0.602 +- 0.052 |
> | ESAN | 0.579 +- 0.035 | 0.605 +- 0.037 |0.581 +- 0.042 | 0.614 +- 0.040 | OOM | OOM |0.629 +- 0.058 | 0.596 +- 0.061 |
> | DropGNN  | 0.533 +- 0.042 | 0.517 +- 0.039 |0.557 +- 0.045 | 0.571 +- 0.050 | OOM | OOM |0.619 +- 0.063 | 0.593 +- 0.072 |
> | UGC-GNN (ours) | $\textbf{0.624}$ +- $\textbf{0.036}$| $\textbf{0.713}$ +- $\textbf{0.022}$ | $\textbf{0.701}$ +- $\textbf{0.025}$ | $\textbf{0.689}$ +- $\textbf{0.019}$ | $\textbf{0.714}$ +- $\textbf{0.011}$ | $\textbf{0.701}$ +- $\textbf{0.032}$ | $\textbf{0.648}$ +- $\textbf{0.033}$ | $\textbf{0.625}$ +- $\textbf{0.060}$ |
>
>
> The supplementary results are very similar to expeiments in our paper (Table 2, Table3, Table4 in the paper). When the universal graph canonization does not exist nor tractable, general graph canonization (GC-GNN) helps to enhance GNNs and can achieve highly competitive results in many scenarios like TU datasets. 2) When the universal graph canonization exists and is tractable Lemma 3.3, GNN enhanced by universal graph canonization (UGC-GNN) successfully achieve the SOTA performance. We will add relevant discussion and supplementary experimental results in the revision.
>
>
> [1] Weisfeiler and leman go neural: Higher-order graph neural networks, AAAI 2019.
>
> [2] Provably powerful graph networks, NeurIPS 2019.
>
> [3] Nested graph neural networks, NeurIPS 2021.
>
> [4] Equivariant Subgraph Aggregation Networks, ICLR 2022.
>
> [5] Papp, P. A., Martinkus, K., Faber, L., & Wattenhofer, R. (2021). DropGNN: Random dropouts increase the expressiveness of graph neural networks. Advances in Neural Information Processing Systems, 34, 21997-22009.
>
> [6] Papp, P. A., & Wattenhofer, R. (2022, June). A theoretical comparison of graph neural network extensions. In International Conference on Machine Learning (pp. 17323-17345). PMLR.

---

> ### Author Response · Authors · 2023-11-20
> **Author Response 2/2**
>
> $\textbf{Q2:}$  Lack of computation cost comparison
>
> $\textbf{Re2:}$ We agree that the comparison of computation cost can emphasize the advantage of graph-canonization-enhanced GNNs in complexity.  (1) For high-order GNNs, PPGN has a space complexity of $\mathcal{O}(n^2)$ for each layer, while 1-2-3 GNNs has a space complexity of $\mathcal{O}(n^3)$ for each layer. Then, as Table 2 in the response 1 shows, 1-2-3 GNNs always meet the OOM (out-of-memory) problem on large-scale datasets (Mayo, RosMap, Cancer). On the other hand, by reducing the embedding dimension and number of layers, PPGN can avoid OOM problem on these datasets as Table 3 in our paper indicates; (2) For subgraph-based GNNs like NGNN (nested GNN), we also implement the similar strategy to restrict the radius of subgraphs to be smaller than 2, then NGNN can avoid the OOM problem, too (Table 3 in our paper). Consequently, here we compare the average computation time per epoch of UGC-GNN against NGNN and PPGN on large-scale datasets. First, we find that UGC-GNN significantly reduces the computation complexity compared to high-order GNN PPGN, and this observation can be explained by the  $\mathcal{O}(n^3)$ time complexity of PPGN and the large size of graphs. On the other hand, NGNN suffers from large subgraph sizes caused by the large average node degree in these datasets, then the extra computation burden is also larger than popular graph datasets with small average node degree (like OGB and TU datasets).
>
>
> ## Table 3. Average computation time per epoch
> |Dataset| Mayo | RosMap | Cancer |
> | :------:| :------: | :------: | :------: |
> | NGNN | 48s | 42s | 334s |
> | PPGN | 143s | 126s | 1261s |
> | UNGNN-GIN | $\textbf{13s}$ | $\textbf{10s}$ | $\textbf{89s}$ |
>
> $\textbf{Q3:}$  GC-GNN doesn't consistently enhance the performance of the GNN backbone. It would be beneficial to delve deeper into which graph properties might influence this fluctuation in performance when integrating the GNN backbone with GC-GNN or UGC-GNN.
>
> $\textbf{Re3:}$ This is a very insightful question. In fact, experimental results (Table3, Table4, Table 5) indicate that UGC-GNN can consistently achieve the SOTA performance. We run multiple experiments to fine-tune hyper-parameters of GC-GNN on OGB datasets, and the following table updates results when base GNN is GIN and dataset is ogbg-molhiv.  Based on these updated results, we find that though GC-GNN can still improve the performance on these two OGB datasets, it is not comparable to the SOTA model. Thus, our next research is to delve deeper into this problem to extend the idea of universal graph canonization to general graph learning problems.
>
>
> |Dataset| ogbg-molhiv | ogbg-molpcba |
> | :------:| :------: | :------: |
> | Base GIN |0.7744 +- 0.0098 | 0.2703 +- 0.0023 |
> |GC-GNN (GIN)| 0.7785 +- 0.0195 | 0.2761 +- 0.0043 |
> | Base GCN |0.7501 +- 0.0140 |0.2422 +- 0.0034 |
> |GC-GNN (GCN)| 0.7609 +- 0.0158 | 0.2510 +- 0.0047 |
>
> $\textbf{Q4:}$   Finding an injective function $l(v|\mathbb{G})$ might be challenging for general graph learning contexts. Furthermore, if there could be multiple choices of $l(v|\mathbb{G})$, it is unclear what the impact of applying different choices of  is on the performance of the proposed model.
>
> $\textbf{Re4:}$ This question is very relevant to the last question (Q3). At this moment, our method of finding universal graph canonization is a general technique applicable to a wide range of real-world graph learning problems. For general graph learning contexts, our paper shows that finding the universal graph canonical labels  $l(v|\mathbb{G})$ can be NP-hard. The theoretical results and experiments in our paper indicates that once a (learnable) node labelling/colouring (discrete or continuous) can be equivalent to canonical labels in breaking node asymmetry among isomorphic graphs, while maintaining the GNNs’ stability,  GNNs’ capability in graph-level prediction can be significantly enhanced. Hence, we believe the next step is to design deep neural architectures to learn such node labelling/colouring, instead of directly solving the NP-hard problem.
>
> If there are multiple choices of universal graph canonization $l(v|\mathbb{G})$, we can arbitrarily choose one as they all can maximize the expressivity and maintain the stability. A very simple example is that if we randomly permute $l(v|\mathbb{G})$ for gene network datasets, the output node colouring is still a universal graph canonization. The following table demonstrates the experimental results.
>
> |Dataset| Mayo | Mayo | RosMap | RosMap | Cancer | Cancer |
> | :------:| :------: | :------: | :------: | :------: | :------: | :------: |
> |Metric | Accuracy |F1 score| Accuracy |F1 score | Accuracy |F1 score |
> | UGC-GNN | 0.624 +- 0.036| 0.713 +- 0.022 | 0.701 +- 0.025 | 0.689+- 0.019 |0.714 +- 0.011 | 0.701 +- 0.032 |
> | UGC-GNN (permute) | 0.624 +- 0.035| 0.712 +- 0.022 | 0.701 +- 0.025 | 0.689+- 0.019 |0.715 +- 0.010 | 0.701 +- 0.032 |

---

> ### Author Response · Authors · 2023-11-29
> **A friendly reminder for discussion**
>
> Dear Reviewer RfLx,
>
> We hope this message finds you well. The author response period for this submission was extended until the end of December 1st as the paper did not have three reviews.
>
> It is close to the extension and we have not yet received feedback from you. Ensuring that our response effectively addresses your concerns is a priority for us. Therefore, might we inquire if you have any additional questions or concerns?
>
> We appreciate your time and dedication committed to evaluating our work.
>
> Best Regards,
>
> The Authors of Submission 9229

---

### Official Review · Reviewer_abzL · 2023-11-01

**Soundness:** 2 fair
**Presentation:** 2 fair
**Contribution:** 1 poor
**Rating:** 6
**Confidence:** 3

**Summary:**

Authors propose to tackle GNN expressivity issues by graph canonization. Either using ordering of nodes implied by the referene data distribution (e.g. image or voxel datasets that can give canonical ordering of nodes) or by running existing fast canonization algorithms. Inclusion of this ordering is shown to improve GNN expressive power and performance on real-world datasets.

**Strengths:**

I find the canonization of graphs to be a very intriguing avenue of making GNNs better or to automatically tailoring generic GNNs for a given data distribution if the canonization is specific for that distribution.

The authors do show that including such canonical order does improve the performance of backbone GNN architectures.

**Weaknesses:**

In general I find the paper quite problematic. Mainly due to the total disregard for related work. Let me split this in three parts:

1) Including ordering of nodes as features is very close to other feature augmentation techniques, widely used for GNN expressive power improvements, such as adding random walk embeddings (https://arxiv.org/pdf/2110.07875.pdf), graph laplacian embeddings (https://arxiv.org/abs/2202.13013 https://arxiv.org/pdf/2201.13410.pdf) or even just random features (https://www.ijcai.org/proceedings/2021/0291.pdf). These are not mentioned among the types of expressive GNN architectures (only higher order and subgraph GNNs are mentioned), while conceptually they are very similar (add pre-computed features to increase power). Neither any of the experiments compare to these existing models.

2) The selection of discussed subgraph-based GNNs is quite peculiar, as it skips in my opinion some of the most popular works in this context (such as DropGNN https://arxiv.org/pdf/2111.06283.pdf and ESAN https://arxiv.org/pdf/2110.02910.pdf as well as various follow up works to these of which there has been many). They also say that subgraph GNNs are not more expressive than 2-WL, which in general is definitely not true, for example see theorem 6.5 in https://proceedings.mlr.press/v162/papp22a/papp22a.pdf where the marking GNN is very closely related to the DropGNN and one of the ESAN variants mentioned above. Also as shown in Table 1 here https://openreview.net/pdf?id=8WTAh0tj2jC experimentally many of these expresive GNNs such as ESAN or DropGNN distinguish the usual 2-WL counter example graphs. (Note that here by 2-WL following the notation in those papers I mean the folklore WL hierarchy, where 2-WL is equal to the 3-WL in the usual WL hierarchy).

3) The known results for the power of assigning unique identifiers to nodes, such as the seminal work of https://arxiv.org/pdf/1907.03199.pdf which shows that GNNs with unique IDs are universal, although generally we don't have a way to assign such IDs in a stable manner. In my eyes the UG-GNN in Tables 4 and 5 essentially just a direct application of this in cases where the data comes with a good ID assignment due to the nature of data distribution.

Note that many more of works in all 3 categories exist, I just cited a few important ones that came to my mind as an example.

Now moving on to the experiments, its again very strange that none of the aforementioned baselines, especially the pre-computed feature ones are included. And generally almost. no expressive GNNs from the past couple of years are included in the benchmarks (e.g. ESAN, CIN, etc.). Not even giving nodes random noise/random IDs is compared against, which is a very usual expressive GNN baseline. Actually it is interesting that in Table 1 the GC-GNN does not achieve 100% test accuracy in any of the cases, usually expressive GNNs manage to hit 100% or at least 99%, with GNN augmented with random features also managing >90% test set accuracy. Random features are usually underperforming other expressive GNNs due to the poor stability of feature (or random ID) assignment, but as the authors point out their GC-GNN can also suffer from poor ordering stability.

So now, for me its completely unclear if this proposed expresivity enhancement is any better than existing, even simplest ones.

While OGB benchmarks are mentioned, the model is not evaluated on any of them? Why is that? It is now the main standard GNN benchmark as the TU-Datasets usually have quite few graphs in them, which can induce high variance of the results.

**Questions:**

Please address the weaknesses above.

---

> ### Author Response · Authors · 2023-11-20
> **Author Response 1/3**
>
> We appreciate the reviewer’s effort in reviewing our paper. The reviewer’s suggestion to compare the proposed method with more expressive GNNs can help to better evaluate the significance and contribution of our work. However, there are some misunderstandings of the methods and experiments in our paper.
>
>
> $\textbf{Q1:}$  Comparison of our method against suggested expressive GNN baselines:  SignNet [1], GNN-RNI[2] (which exists in our paper),  ESAN [3] and DropGNN [4]. This is a question summarized from weakness 1, weakness 2 and other concerns of reviewer abzL.
>
> $\textbf{Re1:}$ Reviewer abzL listed several expressive GNNs for comparison, and they can be categorized into two types: 1) feature augmentation GNNs that adds pre-computed/learnable features to improve the expressivity, such as SignNet [1] and GNN-RNI [2];   2) GNNs with markings that slightly perturbs the input graphs for multiple times to execute GNNs and then aggregates the final embeddings of these perturbed graphs in each run, including ESAN [3] and DropGNN [4].
>
>
> These suggested GNNs are very good baselines, thus we provide the supplementary experiments to compare our (universal) graph canonization enhanced GNN ( GC-GNN/UGC-GNN) against these expressive GNNs.. The additional experiments are also added in the revision.
>
>
> ## Table 1. Supplementary experiments on TU datasets
> |Dataset| DD | MUTAG | PROTEINS | PTC-MR | ENZYMES |
> | :------:| :------: | :------: | :------: | :------: | :------: |
> | SignNet|78.2 +- 4.1 | 88.3 +- 9.2 |75.6 +- 4.1 | 64.3 +- 7.1 | 37.5 +- 6.4 |
> | GNN-RNI| 75.8+- 3.0 | 90.4 +- 7.4 | 73.5 +- 4.5 | 58.2 +- 6.3| 30.7 +- 5.6 |
> |ESAN| 79.7 +- 3.8| $\textbf{91.0}$ +- $\textbf{4.8}$ |75.8 +- 4.5| 65.7 +- 7.1 | 37.9 +- 6.3 |
> | DropGNN| 78.5+- 6.0| 90.4 +- 7.0 |76.3 +- 6.1 | 62.7 +- 8.4 | $\textbf{38.3}$ +- $\textbf{7.1}$ |
> | GC-GNN (ours) |$\textbf{91.3}$ +- $\textbf{9.7}$|86.2 +- 9.9 | $\textbf{76.7}$ +- $\textbf{5.1}$ | $\textbf{66.9}$+- $\textbf{7.1}$ |37.7 +- 6.9 |
>
>
> ## Table 2.  Supplementary experimental results on bioinformatical datasets
> |Dataset| Mayo | Mayo | RosMap | RosMap | Cancer | Cancer |  fMRI-ABIDE | fMRI-ABIDE |
> | :------:| :------: | :------: | :------: | :------: | :------: | :------: |  :------: | :------: |
> |Metric | Accuracy |F1 score| Accuracy |F1 score | Accuracy |F1 score | Accuracy |F1 score |
> | SignNet | 0.527 +- 0.035 | 0.514 +- 0.029 |0.544 +- 0.030 | 0.609 +- 0.037 | 0.572 +- 0.044 | 0.553 +- 0.048 | 0.628 +- 0.041 | 0.597 +- 0.052 |
> | GNN-RNI | 0.513 +- 0.027 | 0.501 +- 0.031 | 0.496 +- 0.041 | 0.512 +- 0.037 | 0.521 +- 0.041 | 0.502 +- 0.069 |0.624 +- 0.021 | 0.598 +- 0.050 |
> | ESAN | 0.579 +- 0.035 | 0.605 +- 0.037 |0.581 +- 0.042 | 0.614 +- 0.040 | OOM | OOM |0.629 +- 0.058 | 0.596 +- 0.061 |
> | DropGNN  | 0.533 +- 0.042 | 0.517 +- 0.039 |0.557 +- 0.045 | 0.571 +- 0.050 | OOM | OOM |0.619 +- 0.063 | 0.593 +- 0.072 |
> | UGC-GNN (ours) | $\textbf{0.624}$ +- $\textbf{0.036}$| $\textbf{0.713}$ +- $\textbf{0.022}$ | $\textbf{0.701}$ +- $\textbf{0.025}$ | $\textbf{0.689}$+- $\textbf{0.019}$ | $\textbf{0.714}$ +- $\textbf{0.011}$ | $\textbf{0.701}$ +- $\textbf{0.032}$ | $\textbf{0.648}$ +- $\textbf{0.033}$ | $\textbf{0.625}$ +- $\textbf{0.060}$ |
>
>
> In the experiment, ESAN implements the stochastic sampling on large-size graphs (i.e. bioinformatical datasets) and uses the ED policy for a strictly higher expressivity than 3-WL in distinguishing SR graphs. DropGNN sets the number of runs to be $k = ceil(\frac{m}{100})$ and dropout probability to be $\frac{1}{k}$, where $m$ is the average graph size. When the model  exceeds the GPU memory, we have an out-of-memory OOM problem.
>
> Table 1 and Table 2 in the response present the supplementary experimental results. The supplementary results are very similar to what is presented in our paper (Table 2, Table3, Table4 in the paper). That is, 1) when the universal graph canonization does not exist nor tractable, general graph canonization (GC-GNN) helps to enhance GNNs and can achieve highly competitive results in many scenarios like TU datasets. 2) When the universal graph canonization exists and is tractable by Definition 3.1 and Lemma 3.3 in our paper, GNN enhanced by universal graph canonization (UGC-GNN) successfully achieve the SOTA performance, as UGC-GNN maximizes the expressivity while maintaining the stability.
>
> [1] Sign and basis invariant networks for spectral graph representation learning
>
> [2] The surprising power of graph neural networks with random node initialization
>
> [3]  Equivariant subgraph aggregation networks.
>
> [4] DropGNN: Random dropouts increase the expressiveness of graph neural networks

---

> ### Author Response · Authors · 2023-11-20
> **Author Response 2/3**
>
> $\textbf{Q2:}$  Weakness 2 states that “The selection of discussed subgraph-based GNNs is quite peculiar, as it skips in my opinion some of the most popular works in this context (DropGNN and ESAN). They also say that subgraph GNNs are not more expressive than 2-WL, which in general is definitely not true.”
>
> $\textbf{Re2:}$ Based on the last paragraph in the first page of our paper, the subgraph-based GNNs are defined as methods that improve the expressivity by encoding (learnable) local structures (i.e. induced k-hop neighborhoods) of nodes. In other words, subgraph-based GNNs formulate hierarchies of local isomorphism on neighborhood subgraphs. Thus, DropGNN and ESAN suggested by the reviewer abzL do not belong to subgraph-based GNNs. In fact, they are GNNs with markings that slightly perturbs the input graphs for multiple times to execute GNNs and then aggregates the learnt final embeddings of these perturbed graphs in each run. As theorem 6.3 in [5] shows, there always exists graphs distinguishable by 2-WL (3-WL) yet not distinguishable by subgraph-based GNNs, then subgraph-based GNNs can not be superior than 2-WL.
>
> For subgraph-based GNNs, our paper have included many well-adopted powerful subgraph-based GNNs, including NGNN, GraphSNN, GNN-AK+.
>
> [5] A theoretical comparison of graph neural network extensions.
>
> $\textbf{Q3:}$  Weakness 3 states that “GNNs with unique IDs are universal, although generally we don't have a way to assign such IDs in a stable manner. In my eyes the UG-GNN in Tables 4 and 5 essentially just a direct application of this in cases where the data comes with a good ID assignment due to the nature of data distribution.”
>
> $\textbf{Re3:}$ Canonical labels generated by graph canonization can be viewed as the node’s ID or position. However, we do not agree that Tables 3, 4 and 5 are essentially a direct application to datasets of good ID assignment. In practice, dominant graph canonization tools like Nauty [6] and Bliss [7] follow the paradigm of individualization and refinement (IR) to obtain the canonical labels, thus we can obtain different canonical labels by using different backtracking search heuristics when exploring the search space. Thus, it is not clear what node ID assignment is ‘good’. Then our theoretical results help to answer this problem: Lemma 2.4 indicates that any canonical labels generated by the IR paradigm is not good in terms of model stability, while Lemma 3.3 proposes methods to find the good ID assignments  (universal graph canonization in Definition 3.1) for many scenarios of great practical importance (which are listed as application scenarios in our paper). Our experimental results in Table3,4,5 align well with these theoretical findings, and Table 3 illustrates that UGC-GNN (universal graph canonization found by Lemma 3.3) significantly outperforms GC-GNN (general graph canonization follows the IR paradigm), indicating that our Lemma 3.3 helps to find 'good' ID assignments in these scenarios.
>
> One of the main contributions of our paper is to show that  the model stability is also important when graph canonization techniques are used to enhance GNNs. In other words, our theory shows that a ‘good’ node ID assignment should be equivalent to canonical labels in breaking node asymmetry among isomorphic graphs, while maintaining the GNNs’ stability.
>
>
> [6] Brendan D McKay and Adolfo Piperno. Practical graph isomorphism, ii. Journal of symbolic computation, 60:94–112, 2014.
>
> [7]Tommi Junttila and Petteri Kaski. bliss: A tool for computing automorphism groups and canonical labelings of graphs. URL http://www. tcs. hut. fi/Software/bliss, 2012.
>
> $\textbf{Q4:}$  Reviewer  abzL states that “no expressive GNNs from the past couple of years are included in the benchmarks (e.g. ESAN, CIN, etc.). Not even giving nodes random noise/random IDs is compared against, which is a very usual expressive GNN baseline”.
>
> $\textbf{Re4:}$ These comments are not true. (1) GNN-RNI (random node initialization)  is already used as a baseline for comparison in Table 3 in our paper to show that the universal graph canonization helps GNN to achieve SOTA performance since it addresses the concern of GNN stability. (2) Baseline GNNs used in our paper include many well-adopted expressive GNN, including popular subgraph-based GNNs (i.e. GraphSNN (2022) ,GNN-AK+(2021), NGNN (2021)), state-of-the-art DAG GNNs like DAGNN, and dominant graph Transformers (i.e. Graphormer (2021), SAN (2021)). These baseline GNNs are widely adopted and have achieved top positions in leaderboards. The reason why we did not include GNNs with markings as baselines in our paper is that they are likely to have the out-of-memory (OOM) problem when applied to large-scale graphs like gene networks in Table 3 (Mayo, RosMap, Cancer). In order to avoid the OOM problem, DropGNN and ESAN are shallow (number layers < 3) and the embedding dimensions are small ($\leq$ 8). Details of datasets are available in the Appendix A.

---

> ### Author Response · Authors · 2023-11-20
> **Author Response 3/3**
>
> $\textbf{Q5:}$  Reviewer abzL states that “Actually it is interesting that in Table 1 the GC-GNN does not achieve 100% test accuracy in any of the cases, usually expressive GNNs manage to hit 100% or at least 99%.” and asks why GC-GNN is not tested on OGB datasets.
>
> $\textbf{Re5:}$ The performance of GC-GNN in Table 1 aligns well with our theoretical findings. As we can see, there always exists a significant gap between the training performance and testing performance of GC-GNN on synthetic datasets. This performance can be explained by Lemma 2.4: the canonical node labeling/colourings used in GC-GNN are generated by graph canonization tools that follow the individualization-refinement paradigm (like Nauty), then GC-GNN is not stable and the generalization ability can not be guaranteed. Consequently, we test GC-GNN on some real-world datasets: TU datasets and OGB datasets. On TU datasets (Table 2 in the paper and Table 1 in the response 1), GC-GNN can achieve highly competitive (even the SOTA) results against other expressive GNNs. However, on OGB datasets, GC-GNN can still improve the performance yet it is not comparable to the SOTA results. These results explain why we formulate the universal graph canonization and design methods to find it (Table 3, 4, 5 in the paper). Now, our method of utilizing universal graph canonization is a general technique applicable to a wide range of real-world problems, and our next research is to extend this method to all graph learning problem.
>
>
> In the paper, experiments on OGB datasets are in Table 7 (see Appendix B), and we describe the experimental results in the start of section 4.4. Here we summarize the results, and compare it with ESAN. When GIN is used as a base model in ogbg-molhiv, we fine-tune hyper-parameters of GC-GNN (GIN) to get better results than Table 7. Similar to ESAN, GC-GNN can improve the base GNN model, yet can not achieve the state-of-the-art performance.
>
> ## Table. Experimental results on OGB (Table 7 in our paper)
> |Dataset| ogbg-molhiv | ogbg-molpcba |
> | :------:| :------: | :------: |
> | Base GIN |0.7744 +- 0.0098 | 0.2703 +- 0.0023 |
> | ESAN (GIN + ED)| 0.7803 +- 0.0170 | 0.2782 +- 0.0036 |
> |GC-GNN (GIN)| 0.7785 +- 0.0195 | 0.2761 +- 0.0043 |
> | Base GCN |0.7501 +- 0.0140 |0.2422 +- 0.0034 |
> | ESAN (GCN +ED)| 0.7559 +- 0.0120 | 0.2521 +- 0.0040 |
> |GC-GNN (GCN)| 0.7609 +- 0.0158 | 0.2510 +- 0.0047 |

---

> ### Author Response · Authors · 2023-11-29
> **A friendly reminder for discussion**
>
> Dear Reviewer abzL,
>
> We hope this message finds you well. The author response period for this submission was extended until the end of December 1st as the paper did not have three reviews.
>
> It is close to the extension and we have not yet received feedback from you. Ensuring that our response effectively addresses your concerns is a priority for us. Therefore, might we inquire if you have any additional questions or concerns?
>
> We appreciate your time and dedication committed to evaluating our work.
>
> Best Regards,
>
> The Authors of Submission 9229

---

> ### Comment · Reviewer_abzL · 2023-11-30
> **Response to the Rebuttal**
>
> I thank the authors for their response and apologise for my late response.
>
> Regarding subgraph GNNs: its true that DropGNN can be understood through the marking prism, but ESAN goes beyond that.
> For example the EGO and EGO+ strategies from ESAN, build ego graphs (potentially with root being marked) for each node and certainly share similarities to the strategy presented in "Nested Graph Neural Networks" by Zhang and Li that you reference as a subgraph model example. At the end of the day ESAN has Subgraph literally in the name and follow-up works do classify it as a subgraph GNN (e.g. https://proceedings.neurips.cc/paper_files/paper/2022/file/cb2a4cc70db72ea779abd01107782c7b-Paper-Conference.pdf). So if you don't mean to include it in your definition of subgraph GNN, this should be made clear (the two versions compared) and the separation boundary rigorously specified.
>
> I appreciate the extra experiments you have performed.
>
> I'm still not convinced by the significance of the contribution of this work. In the sense that it does not extend much beyond the current knowledge present in the field (good IDs are good).
> It is quite unfortunate that we are the only two reviewers, so accounting for that I still raise my score to a week accept, but lower my confidence in this.
>
> The PDF seems to not have been updated with the additional results. Why is that? It would be nice to see the final picture of the work.
> But I trust that all the changes discussed will be incorporated in the final version.
>
> By the way, a very common expressive GNN benchmark is the ZINC dataset, which discriminates the different expressive architectures quite well, while for example TU-Datasets are quite murky (most models do similarly). It would make the final paper version stronger if this is included.

---

> ### Author Response · Authors · 2023-12-02
>
> Again, we would like to express our deep gratitude for these valuable comments, and they are helpful to improve our paper.
>
> $\textbf{Q1:}$ Discussion regarding to subgraph-based GNNs.
>
> $\textbf{Re1:}$ We agree that ESAN with EGO and EGO+ policy can be considered as a subgraph-based GNN. However, when the node-deleted subgraphs (ND) and the edge-deleted subgraphs (ED) polices are used, a graph is mapped to the set of subgraphs obtained  by removing a single node or edge, then it can be understood through the marking prism. Thus, it is really a good suggestion to clarify this point to avoid ambiguity.
>
> $\textbf{Q2:}$ Response to 'In the sense that it does not extend much beyond the current knowledge present in the field (good IDs are good).'
>
> $\textbf{Re2:}$. In the revision, we summarize that current dominant expressive GNNs as: high-order GNNs, subgraph-based GNNs, GNNs with marking, and feature augmentation GNNs. Overall, these expressive GNNs are powerful, yet still far from efficiently distinguishing any pair of nonisomorphic graphs. For instance, high-order GNNs and GNNs with markings are not scalable to large-scale graphs due to the space and time complexity; subgraph-based GNNs have an upper bounded expressicity as there always exists graphs distinguishable by 2-WL (3-WL) yet not distinguished by subgraph-based GNNs; Powerful feature augmentation GNNs like SignNet is also dependent on eigendecomposition, thus not suitable for large-scale graphs. Consequently, we propose to use graph canonization to enhance GNNs, as GNNs with graph canonization techniques (i.e. GC-GNN and UGC-GNN) can maximize the expressivity while do not increase the space/time complexity compare to a standard MP-GNN (message passing GNN).
>
> Though we know that 'good IDs are good', it is unclear what IDs are good in the field of graph canonization, and any IDs generated by graph canonization tools can help to maximize the expressivity of GNNs. Thus, our paper propose to study what IDs are good in the field of graph canonization from the perspective of GNN's stability.
>
> $\textbf{Q3:}$ PDF seems to not have been updated with the additional results.
>
> $\textbf{Re3:}$. We just update the PDF (rebuttal revision) and it should be visible now. We were waiting for the third reviewers' comments thus we did not update it. Based the feedback, ACs will take a closer look at the paper if the third reviewer is missed. Thus, we prepare the revision based on available comments from current two reviewers.
>
> $\textbf{Q4:}$ Discussion of ZINC dataset.
>
> $\textbf{Re4:}$. We agree that ZINC dataset is a good expressive benchmark dataset. As we pointed out in the Appendix B, a major limitation of graph canonization is that it cannot manipulate the heterogeneity of edges. In ZINC datasets, graphs contain edges of different types, and typical (expressive) GNNs usually uses an edge encoder (nn.Embedding(4, emb_dim)) for the edge embedding to encode the heterogeneity. Thus, these graphs are essentially not ideal to test graph-canonization-based method.
>
> However, we tried to integrated our method of using node IDs from graph canonization as augmented features in subgraph-based GNNs (NGNN). That is, for each subgraph extracted in NGNN, we run GNNs with graph canonization to obtain the subgraph embedding. In this case, our method can achieve a test MAE of $0.093 \pm 0.005$, which is a competitive result.

---

### Meta-Review · Area_Chair_va1y · 2023-12-20

**Metareview:**

All reviewers appreciate the novelty of the idea and the theoretical result. There were initial concerns including discussion with respect to prior works and empirical study, but were fixed later by the author. Both reviewers recommend accept.

**Justification For Why Not Higher Score:**

The significance of the contribution could be further clarified or developed.

**Justification For Why Not Lower Score:**

This paper provides an interesting solution to a timely problem, and provides a solid empirical study.

---

### Decision · Program_Chairs · 2024-01-16

Accept (poster)